# Meta-analysis of the impacts of global change factors on soil microbial diversity and functionality

Zhenghu Zhou [1,2], Chuankuan Wang [1,2✉] & Yiqi Luo[3]

Biodiversity on the Earth is changing at an unprecedented rate due to a variety of global change factors (GCFs). However, the effects of GCFs on microbial diversity is unclear despite that soil microorganisms play a critical role in biogeochemical cycling. Here, we synthesize 1235 GCF observations worldwide and show that microbial rare species are more sensitive to GCFs than common species, while GCFs do not always lead to a reduction in microbial diversity. GCFs-induced shifts in microbial alpha diversity can be predominately explained by the changed soil pH. In addition, GCF impacts on soil functionality are explained by microbial community structure and biomass rather than the alpha diversity. Altogether, our findings of GCF impacts on microbial diversity are fundamentally different from previous knowledge for well-studied plant and animal communities, and are crucial to policy-making for the conservation of microbial diversity hotspots under global changes.

[1] Center for Ecological Research, Northeast Forestry University, Harbin 150040, China. [2] Key Laboratory of Sustainable Forest Ecosystem Management-Ministry of Education, Northeast Forestry University, Harbin 150040, China. [3] Center for Ecosystem Science and Society, Northern Arizona University, Flagstaff, AZ 86011, USA. ✉email: wangck-cf@nefu.edu.cn

Human-induced global change factors (GCFs), such as climate warming (W), carbon-dioxide enrichment (eCO$_2$), altered precipitation, atmospheric nitrogen (N) deposition, nutrient fertilization, land-use change (LUC), and their combinations, seriously threaten the biodiversity in our planet[1,2]. LUC is the dominant driver of biodiversity decline in terrestrial ecosystems mainly through loss, degradation, and fragmentation of the habitats[1,3]. Projected climate changes cause species extinction once the species falls outside its climatic niche[4]; and 15–37% of species are expected to go extinct under the mid-range scenarios of temperature and CO$_2$ rises[5]. Enhanced atmospheric N deposition leads to changes in plant species interaction and community composition, results in soil acidification and ion toxicity, decreases the resistance of plants to pathogens and insect pests, and consequently is recognized as the third greatest driver (after LUC and climate changes) of biodiversity loss in the century[1,6]. Soil microbial communities play a critical role in almost all of the biogeochemical cycling processes in terrestrial ecosystems, such as organic matter decomposition, nutrient cycling, plant diversity, and productivity[7–9]. However, our understanding of how GCFs affect the biodiversity and its relations to the functionality for microorganism lags substantially behind that for macroorganisms (plants and animals)[1–6]. These knowledge gaps swamp our predictions of GCFs impacts on microbial diversity and thus constrain the establishment of effective policies to preserve microbial diversity hotspots.

Soil microbial communities are surprisingly diverse and abundant[10]. It has been estimated that 1 trillion (10$^{12}$) microbial species harbor on the Earth[10], and 1 g soil contains up to 1 billion (10$^9$) bacterial cells consisting of tens of thousands of taxa, which raise great challenges to investigate microbial diversity[8]. Scientists have attempted to examine whether microbial diversity displays an environmental gradient like plant diversity, and whether microbial community assembly follows the macroecological theories[10–16], such as the metabolic theory, the species energy theory,

the stoichiometry theory, and plant–soil interaction (Supplementary Table 1). Yet, such attempts often fail for soil microorganisms[13–15]. For example, a recent global meta-analysis of 325 soil communities showed that the driver of microbial diversity was often inconsistent among different studies[15]. Despite recent individual experiments have examined the responses of microbial diversity to GCFs, the effect of GCFs on microbial diversity remains highly elusive and inconclusive.

Current evidence of plant community studies supports a positive but saturating relationship between plant biodiversity and ecosystem functioning[17,18], which can result from niche complementarity, positive interactions, greater use of limiting resources, decreased herbivory and pathogens, the presence of certain influential species, etc[17,18]. Consequently, an enormous amount of research claims that ecosystem functioning is threatened by an ongoing loss of species due to GCFs[1–3,19]. A common notion seems to be developed, largely inspired by the studies on aboveground communities, that microbial diversity drives microbial functionality in terrestrial ecosystems[7,8,20,21]. However, soil microorganisms are suggested to be too diverse and abundant to assume that the biogeochemical cycling is limited by the microbial diversity[22–24]. It is still unclear whether the biodiversity loss of microbial communities reduces microbial functionality in the ecosystems under GCFs.

Here, we conducted a global synthesis of 1235 GCF experimental observations that measured microbial alpha diversity (number of species coexisting within a local site), beta diversity (the magnitude of similarity in species composition among different sites), and community structure with high-throughput sequencing techniques, and corresponding biomass and ecosystem functionalities from eight types of biomes (agriculture, tundra, temperate/boreal forest, tropical/subtropical forest, Mediterranean vegetation, grassland, desert, and wetland; Supplementary References, Fig. 1, and Datasets 1–3). The microbial groups include bacteria, fungi, and six specialized microbes (i.e.,

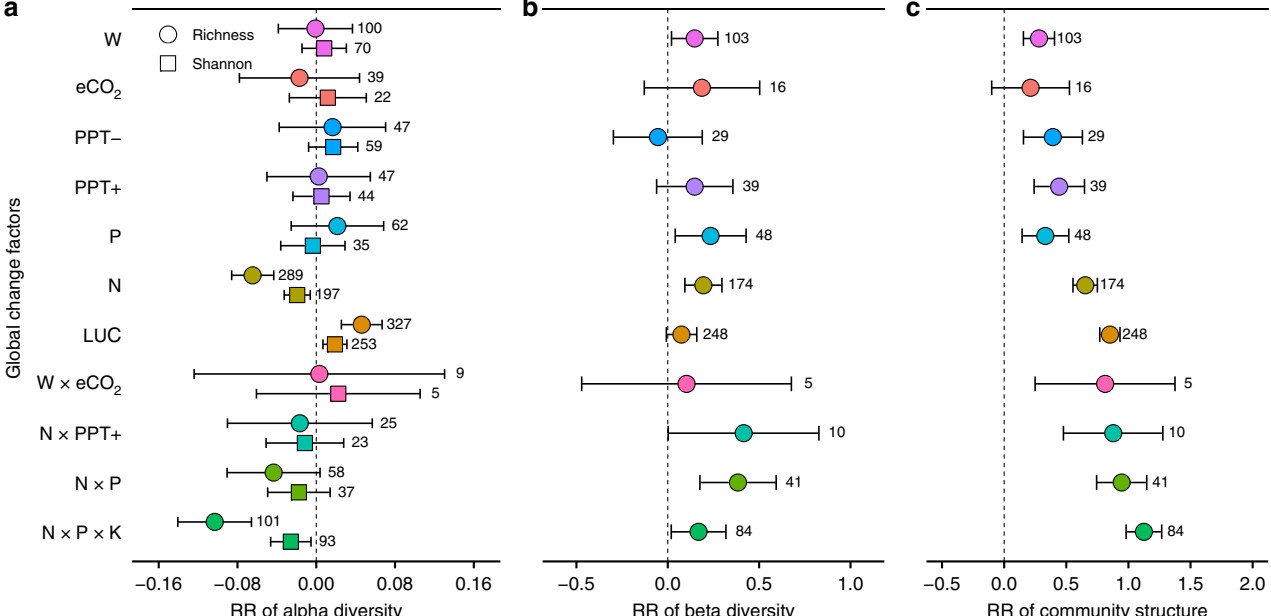

**Fig. 1 Responses of microbial diversity and community structure. a** Response ratio (natural logarithm-transformed ratio of treatment to control, RR) of microbial alpha diversity (richness and Shannon index). **b** RR of microbial beta diversity. **c** RR of microbial community structure. Weighted means and their 95% confidence intervals of RRs are given. The numbers at the right side of the confidence intervals represent the sample sizes. W warming, eCO$_2$ carbon-dioxide enrichment, PPT− decreased precipitation, PPT+ increased precipitation, P phosphorous addition, N nitrogen addition, LUC land-use change, W × eCO$_2$ warming plus carbon-dioxide enrichment, N × PPT+ nitrogen addition plus increased precipitation, N × P nitrogen plus phosphorous addition, N × P × K nitrogen plus phosphorous plus potassium addition. Source data are provided as a Source Data file.

denitrifier, nitrifier, diazotroph, phosphorous (P) mineralizer, methanotroph, and methanogen). The GCFs include seven single-factor experiments (W, $eCO_2$, precipitation addition (PPT+), precipitation reduction (PPT−), N deposition, P addition, and LUC) and four combined-factor experiments (W × $eCO_2$, N × PPT+, N × P addition, and N × P × potassium (N × P × K) addition). The questions we address include: first, what are the effects of GCFs on microbial diversity and community structure worldwide? Are the effects similar to those reported for plants and animals? Second, what are the potential drivers of these responses? As potential drivers of microbial responses to GCFs, climate, biome type, microbial group, soil resources, soil pH, and experimental forcing factors were evaluated by the model selection analysis based on the corrected Akaike information criterion (AIC). Finally, how do GCFs-induced changes in microbial alpha diversity affect the microbial functionality in the ecosystems?

Our results show that microbial community structure is sensitive to GCFs, while GCFs affect microbial diversity inconsistently and do not always lead to the loss of microbial diversity. Conversion from highly diverse natural ecosystems to homogeneous agricultural monocultures has a positive effect on microbial alpha diversity. Soil pH is the most important factor to predict the GCFs effects on microbial alpha diversity. Generally, if a GCF increases the soil pH, the alpha diversity would increase; if it decreases the soil pH, the alpha diversity would reduce; if it has no effect on soil pH, it would not change the alpha diversity. The response of soil functionality to GCFs can be explained by the responses of microbial community structure and biomass rather than the response of microbial alpha diversity. In summary, our findings of GCFs impacts on microorganisms are fundamentally different from previous knowledge for the well-studied plant and animal communities.

## Results and discussion

**GCFs have little negative effect on microbial biodiversity.** Pooling all the data across microbial groups and biomes, we found that GCFs do not always cause microbial diversity loss (Fig. 1a, b) like that for aboveground communities[1–6]. The areas of croplands, pastures, and plantations have been expanded globally in recent decades, accompanied by large losses of the alpha diversity of plants and animals[1,3] and biotic homogenization (beta diversity loss) as well[25]. Surprisingly, LUC has a positive effect on microbial diversity, i.e., a significant increase of alpha diversity and an insignificant positive effect on beta diversity (Fig. 1a, b and Supplementary Fig. 2); such a positive effect still exists when the paired alpha diversity and beta diversity from the same case studies are compared (Supplementary Fig. 3). In addition, a significant increase in alpha diversity during the conversions from highly diverse natural ecosystems to homogeneous agricultural monocultures (Supplementary Fig. 2) implies that changes in microbial alpha diversity are also uncoupled with the shifts in plant alpha diversity. W, $eCO_2$, altered precipitation, and P addition do not result in loss of microbial alpha diversity either, while W and P addition even improve the beta diversity (Fig. 1a, b). N and N × P × K additions significantly decrease the richness and Shannon index but increase the beta diversity (Fig. 1a, b), indicating that their effect on microbial diversity is uncertain. In addition, we found that the absolute values of the response ratios (RRs, natural logarithm-transformed ratio of treatment to control) of richness are consistently greater than those of Shannon index (Fig. 1a), suggesting that microbial rare species are more sensitive to GCFs than common species, in agreement with a previous study[26]. We also found that GCFs greatly change microbial community structure regardless of the effects of GCFs on microbial diversity mentioned above. The RRs of the community structure to all the GCFs (except for $eCO_2$) are significantly greater than zero especially for the

combined factors (Fig. 1c). This suggests that the response of microbial community structure to GCFs is more sensitive than that of microbial diversity, perhaps because of their different drivers (Supplementary Figs. 4–6), consistent with previous works[27].

We further split the data by microbial groups or biome types and aforementioned patterns of GCFs effects on microbial alpha diversity are largely maintained (Fig. 2). The macroecology theory on ecosystem succession and disturbance have been applied in microbial ecology field[28], and LUC decreases the ratio of K-strategical fungi to r-strategical bacteria and this ratio also has a rising trend during secondary succession[29,30]. Fungi and bacteria can be also differentiated into oligotrophic and copiotrophic categories, and the former have higher biomass C to nutrients ratios and need less nutrients[12,31]. LUC-induced shifts in community structure from fungi dominated to bacteria dominated[29] and decreases in soil C:N (Supplementary Fig. 7) may together explain the greater positive response of the alpha diversity of bacteria to LUC than that of fungi (Fig. 2a, b). A previous study had found that N addition increases bacterial growth but inhibits fungal biomass[32]. Nevertheless, fungi and bacteria have comparable negative RRs of richness and Shannon index to N, N × P, and N × P × K additions (Fig. 2a, b). The RR of Shannon index to W is affected by microbial groups (Fig. 2b), but the underlying mechanism is unknown given that W does not change the fungi to bacteria ratio and which microbial group would dominate in the warmed plots is unclear[33,34]. PPT− increases the fungal richness but does not change the bacterial richness, while PPT+ increases the bacterial alpha diversity but does not change the fungal alpha diversity. This phenomenon partly coincides with our expectation because fungi are thought to have a greater capability to tolerate water stress than bacteria due to their ability to accumulate osmoregulatory solutes to protect their metabolism and filamentous structure[35–37]. Specialized microbes show some different responses to GCFs compared with fungi and bacteria, but these differences are inconsistent between richness and Shannon index (Fig. 2a, b).

A positive effect of W on microbial diversity in cold region is anticipated because W has stronger positive effects on microbial growth in cold than warmer regions[34]. However, tundra and temperate/boreal forests have similar RRs of richness and Shannon index to W compared with tropical/subtropical forests (Fig. 2c, d), and RR of microbial alpha diversity to W is also decoupled with mean annual temperature (MAT) (Supplementary Table 2). Microbial biomass is more sensitive (positive response) to PPT+ at xeric than mesic sites, and it is more responsive (negative response) to PPT− in humid than dry sites[38]. Such patterns, however, do not exist for the RR of microbial alpha diversity (Fig. 2c, d), i.e., this RR is comparable between humid and dry biomes (Fig. 2c, d) and no significant correlation between RR of microbial alpha diversity to altered precipitation and mean annual precipitation (MAP) is found (Supplementary Table 2). We also expect a positive effect of P addition on microbial diversity in low latitudes with high MAT and MAP because of the increasing P limitation from geologically young tundra and boreal ecosystems toward tropical forest[39–41], while no significant correlation between RR of microbial alpha diversity and MAT/MAP is found (Supplementary Table 2). Some significant negative relationships exist for GCFs associated with N inputs (N, N × P, and N × P × K additions, and N × PPT+; Supplementary Table 2), which partly due to the high N availability in the tropical regions[39,40], external N input may have stronger negative effects in low latitudes than high latitudes. Similarly, although there are some differences in the RRs of beta diversity and community structure to GCFs among microbial groups and biomes, the potential mechanisms are missing partly due to the limited sample size (Supplementary Fig. 8).

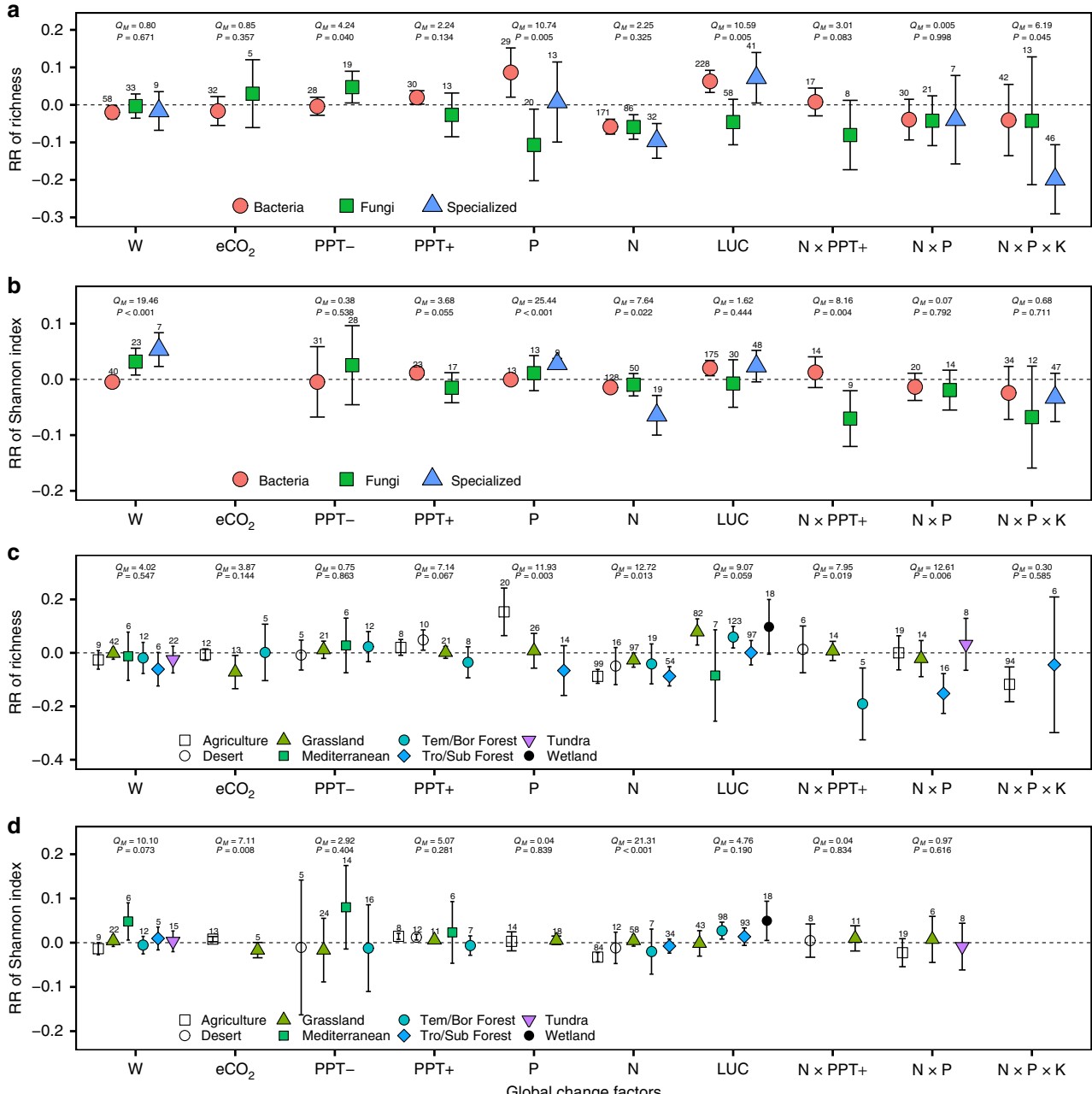

**Fig. 2 Responses of microbial alpha diversity across microbial groups and biomes. a** Response ratio (RR) of richness across microbial groups. **b** RR of Shannon index across microbial groups. **c** RR of richness across biomes. **d** RR of Shannon index across biomes. Weighted means and their 95% confidence intervals of RRs are given. The numbers at the top of the confidence intervals represent the sample sizes. The significances of microbial groups and biome types are tested by the omnibus test ($Q_M$). W warming, $eCO_2$ carbon-dioxide enrichment, PPT− decreased precipitation, PPT+ increased precipitation, P phosphorous addition, N nitrogen addition, LUC land-use change, W × $eCO_2$ warming plus carbon-dioxide enrichment, N × PPT+ nitrogen addition plus increased precipitation, N × P nitrogen plus phosphorous addition, N × P × K nitrogen plus phosphorous plus potassium addition. Tem/Bor temperate/boreal. Tro/Sub tropical/subtropical. Source data are provided as a Source Data file.

Overall, these findings demonstrate that GCFs have little negative effect on microbial biodiversity unlike for aboveground communities[1–6,25], suggesting that the responses of biodiversity to GCFs may be decoupling between above- and below-ground compartments in terrestrial ecosystems.

**Soil pH dominantly controls the alpha diversity responses.** RRs of microbial richness ($R^2 = 0.83$, $P < 0.001$) and Shannon index ($R^2 = 0.81$, $P < 0.001$) significantly increase as the changes in soil

pH increase (Fig. 3b). These correlations between changes in soil pH and RR of microbial alpha diversity maintain across different microbial groups and biome types (Fig. 4). The model selection analysis further indicated that the changed soil pH by GCFs is the most important predictor for the RRs of richness and Shannon index among the potential factors examined, such as soil resource contents and stoichiometry, microbial groups, biomes, climates, and experimental forcing factors (Fig. 3a and Supplementary Figs. 4, 5). As previous findings[11,12], we did observe some positive effects of soil C and N contents and negative effect of soil C:N

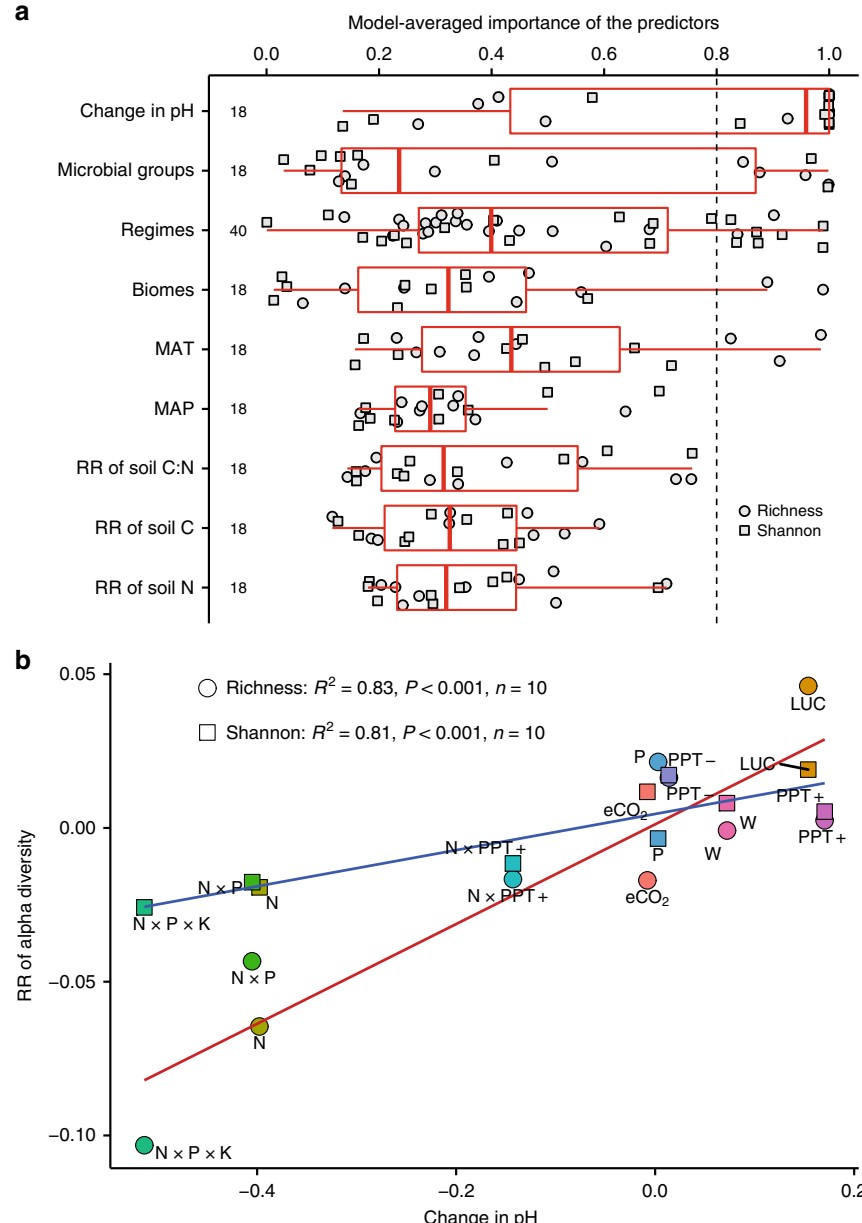

**Fig. 3 Contributors to response of microbial alpha diversity to global change factors. a** Model-averaged importance of the predictors for response ratios (RRs) of richness and Shannon index based on the sum of Akaike weights derived from the model selection using AIC (Akaike's information criteria corrected for small samples). A cutoff of 0.8 (the dashed line) is set to differentiate between important and nonessential predictors. Boxplots show whiskers (denote the lowest and highest values), quartiles, and median of model-averaged importance for different predictors (see Supplementary Fig. 4). **b** Linear relationships between changes in soil pH and RRs of richness ($y = 0.16x + 0.001$) and Shannon index ($y = 0.06x + 0.005$). W warming, $eCO_2$ carbon-dioxide enrichment, PPT− decreased precipitation, PPT+ increased precipitation, P phosphorous addition, N nitrogen addition, LUC land-use change, W × $eCO_2$ warming plus carbon-dioxide enrichment, N × PPT+ nitrogen addition plus increased precipitation, N × P nitrogen plus phosphorous addition, N × P × K nitrogen plus phosphorous plus potassium addition. Source data are provided as a Source Data file.

stoichiometry on microbial alpha diversity (see the weighted averages of model coefficients in Supplementary Fig. 5). Although some correlations exist between changes in soil pH and the RRs of soil resource contents and stoichiometric ratio (Supplementary Fig. 7), the variability in the RR of alpha diversity to GCFs is dominantly explained by the changed soil pH. Specifically, atmospheric N deposition, N × P, and N × P × K additions increase the soil C and nutrient contents and decrease soil C to nutrient ratio (Supplementary Fig. 7), but they have negative effects on microbial richness and Shannon index (Fig. 1a). Conversely, LUC reduces soil C and nutrient contents

(Supplementary Fig. 7), but significantly stimulates microbial alpha diversity (Fig. 1a). In addition, the model-averaged importance for the RRs of soil C, N, and C:N is consistently lower than 0.8 (the cutoff between important and nonessential predictors; Supplementary Fig. 4).

Soil pH is an important predictor for microbial alpha diversity in response to GCFs observed in this synthesis, which is consistent with other studies at local sites[42] and large spatial scales[13,14]. The interpretation is that soil pH plays an important role in membrane-bound proton pumps and protein stability[43], and thus directly imposes a physiological constraint on microorganisms,

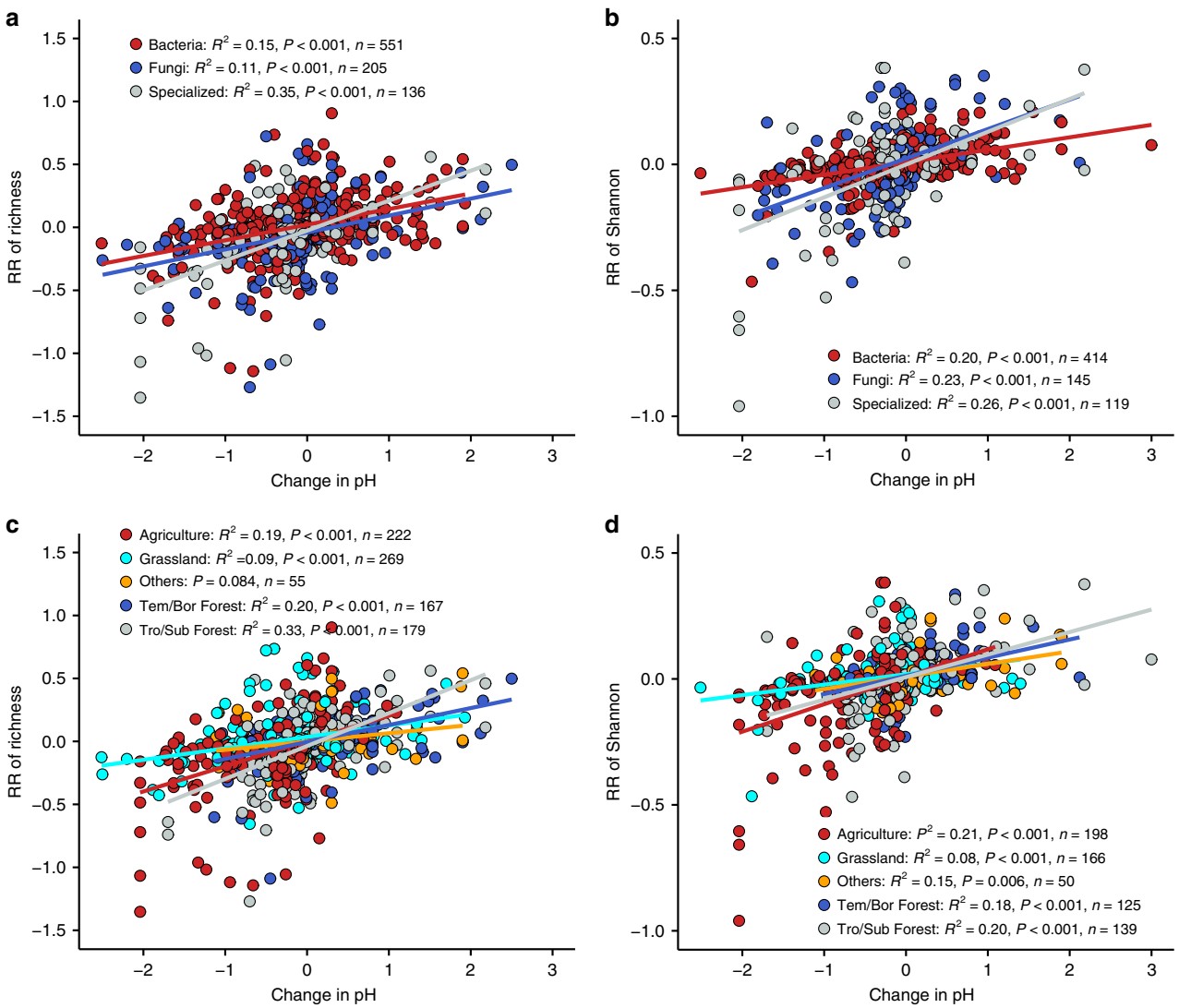

**Fig. 4 Coordinated changes between soil pH and diversity. a** Linear relationships between changes in soil pH and response ratio (RR) of richness by microbial groups (bacteria: $y = 0.12x + 0.02$; fungi: $y = 0.13x - 0.04$; specialized microbes: $y = 0.24x - 0.03$). **b** Linear relationships between changes in soil pH and RR of Shannon index by microbial groups (bacteria: $y = 0.05x + 0.008$; fungi: $y = 0.12x + 0.02$; specialized microbes: $y = 0.13x - 0.0007$). **c** Linear relationships between changes in soil pH and RR of richness by biome types (agriculture: $y = 0.20x + 0.003$; grassland: $y = 0.09x + 0.04$; others: insignificant; temperate/boreal (Tem/Bor) forest: $y = 0.13x - 0.005$; tropical/subtropical (Tro/Sub) forest: $y = 0.26x - 0.04$). **d** Linear relationships between changes in soil pH and RR of Shannon index by biome types (agriculture: $y = 0.11x + 0.009$; grassland: $y = 0.04x + 0.02$; others: $y = 0.05x + 0.01$; Tem/Bor forest: $y = 0.07x + 0.01$; Tro/Sub forest: $y = 0.90x + 0.007$). Source data are provided as a Source Data file.

which reduces the net growth of individual taxa unable to survive when the soil pH falls outside a certain range (niche) and may alter the competitive outcomes. Consequently, extreme pH exerts a significant stress to some taxa that may be less tolerant than others (i.e., alkaliphiles or acidophiles)[13,14,42,43].

**Microbial diversity-structure-biomass-function responses.** LUC reduces both microbial biomass and functionality (including 16 microbial functions related to soil biogeochemical cycling; see "Methods"), while W, PPT+, N × PPT+, and nutrients inputs significantly increase the microbial function, which are different with the response of microbial alpha diversity (Figs. 1a and 5a, b). Consequently, GCFs-induced changes in microbial alpha diversity do not mirror their functionality. Instead, significant and negative relationships are found between RR of microbial functionality and RR of microbial richness ($R^2 = 0.78$, $P < 0.001$) and RR of Shannon index ($R^2 = 0.73$, $P < 0.001$) (Fig. 5c), and the negative or decoupled relationships exist within different

microbial functions associated with decomposition (microbial respiration), net N mineralization rate, oxidative C-cycling enzymes, hydrolytic C-cycling enzymes, N-cycling enzymes, and P-cycling enzymes (Supplementary Fig. 9). In addition, microbial alpha diversity does not mirror microbial biomass production (Fig. 5d). These findings are distinctive from the positive but decelerating richness–functionality relationship in macroecology[17,18]. One potential explanation is that a consortium of microorganisms that carries out soil biogeochemical processes is characterized by a redundancy of functions[24]; and loss of some groups of the species may have little or no effect on overall functionality because other groups can take their place[22–24]. Microbial community structure is sensitive to GCFs (Fig. 1c), and a positive relationship between RR of community structure and functionality is observed ($R^2 = 0.53$, $P = 0.011$; Fig. 5e), implying that variations in microbial community structure might play an important role in the functionality changes despite that it is difficult to tell which

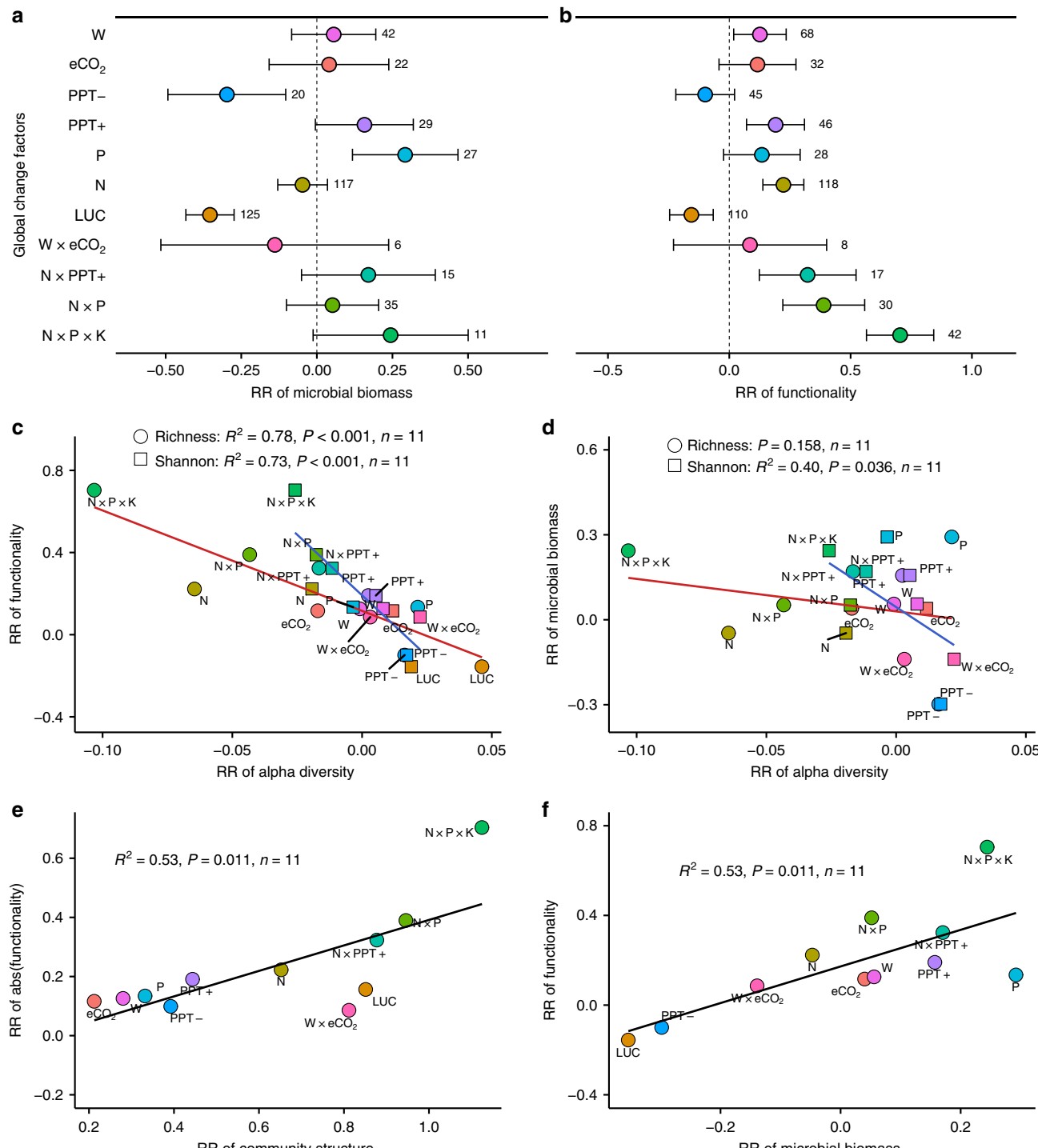

**Fig. 5 Global changes affect microbial biomass and functionality associated with diversity. a** Response ratio (RR) of microbial biomass to global change factors (GCFs). **b** RR of microbial functionality to GCFs. Weighted means and their 95% confidence intervals of RRs are given. The numbers at the right side of the confidence intervals represent the sample sizes. **c** Linear relationships between RR of microbial functionality and RRs of richness ($y = -4.89x + 0.12$) and Shannon index ($y = -11.80x + 0.19$). **d** Linear relationships between RR of microbial biomass and RRs of richness (insignificant) and Shannon index ($y = -7.83x + 0.02$). **e** Linear relationship between RR of absolute value (abs) of microbial functionality and RR of microbial community structure ($y = 0.43x - 0.04$). The absolute value is used because we assume that great change in microbial community structure would result in great change (both increase or decrease) in functionality and the community structure can not be quantified as increasing or decreasing. **f** Linear relationship between RR of microbial functionality and RR of microbial biomass ($y = 0.82x + 0.17$). W warming, eCO$_2$ carbon-dioxide enrichment, PPT− decreased precipitation, PPT+ increased precipitation, P phosphorous addition, N nitrogen addition, LUC land-use change, W × eCO$_2$ warming plus carbon-dioxide enrichment, N × PPT+ nitrogen addition plus increased precipitation, N × P nitrogen plus phosphorous addition, N × P × K nitrogen plus phosphorous plus potassium addition. Source data are provided as a Source Data file.

microbial species shifts determining the changes in the config-uration of samples on the ordination plots.

It is important to note that active alpha diversity rather than total alpha diversity may be positively correlated with the ecosystem functionality[21], and active microorganisms compose only about 0.1–2% of the total microbial biomass[44]. However, cautions should be taken when interpreting this notion. On the one hand, the microbial transition from potentially dormant to active state can occur quickly (in minutes to hours)[44]. The total biomass of microbial community is positively correlated with the microbial functions ($R^2 = 0.53$, $P = 0.011$; Fig. 5f), which suggests that the whole microbial community is important for the functions[22]. On the other hand, positive relationships between active microbial alpha diversity and functions may not be universal[27] because there is similar alpha diversity between total and active microbial communities[45]. In addition, concerns have been raised that not all functions are carried out by the whole microbial community; instead, some key soil functions may be carried out by specialized microbes (like ammonia oxidizer, diazotrophic, methanotrophic, phosphorus mineralizer, etc.), which may be vulnerable to diversity loss partly due to their lower richness[46,47]. Although few studies have evaluated GCFs impacts on both specialized microbial diversities and the functions synchronously in our dataset (Supplementary Data-set 1), the limited data reveal significant negative relationships between RR of alpha diversity and RR of functionality for denitrifier and nitrifier, and decoupled correlations for diazo-trophic and P mineralizer communities (Supplementary Fig. 10). A recent study also found that soil N transformation rates are regulated by the biomass of ammonifiers and nitrifiers but not the alpha diversity[48]. Collectively, the aforementioned results suggest that GCFs-induced changes in soil microbial alpha diversity have no significant effect on the shifts of microbial functionality in the ecosystems.

In summary, our global synthesis verifies that microbial rare species are more sensitive to GCFs than common species and microbial diversity is not always threatened by GCFs. Increase in microbial alpha diversity during LUC implies the decoupling of plant and microbial alpha diversity. Microbial groups, biomes, climates, experimental forcing factors, resource contents, and stoichiometric ratios contribute to the variabilities in RRs of richness and Shannon index, while GCFs shape microbial alpha diversity predominately by changing the soil pH. Microbial alpha diversity is not necessary to mirror microbial biomass production and functionality in the ecosystems, while GCFs-induced shifts in soil functionality are well explained by the changes in microbial community structure and biomass. Overall, our findings indicate that the responses of microbial communities to global changes are fundamentally different from those of macro-communities, which are crucial to the policy-making to preserve microbial diversity hotspots under global environmental changes.

## Methods

**Data collection.** An extensive literature survey was conducted through the Google Scholar databases until February 2020 with no restriction on publication year. The keywords were "soil" and "OTU" and "W or elevated carbon-dioxide or elevated $CO_2$ or $CO_2$ enrichment or drought or decreased precipitation or altered pre-cipitation or increased precipitation or water addition or phosphorus addition or N addition or N deposition or fertilization or land use or forest conversion". A total of 1235 observations of GCF experiments were included in the present study from 341 publications (Supplementary References), which quantified the microbial diversity and community structure by high-throughput sequencing techniques, including the data from Illumina, 454, and ABI platform (Supplementary Fig. 1 and Data-sets 1–3). The following criteria were used to select appropriate studies: (1) only field studies of GCFs are selected, and laboratory incubation studies are not included; (2) at least one microbial community metric, including alpha diversity (OTU, Chao, ACE, or Shannon index), beta diversity, and community structure is

reported; (3) the study duration of the experiment is longer than 1 year/growing season.

Besides the microbial community metrics, the dataset also included soil pH, soil organic C, soil total N, location (i.e., latitude and longitude), MAT, MAP, and experimental forcing factors (i.e., magnitude of W, $CO_2$ concentration elevation, percentage changes in precipitation, addition rate of N or P or K, LUC types (conversion of native ecosystem to secondary ecosystem, plantation, pasture, or agricultural land; secondary ecosystem differs from plantation and pasture mainly in terms of human activity involved in the stand establishment and development[29,49]), and length of the manipulation experiment). Overall, the dataset covered broad variations in ecosystem types, climates, magnitude of GCFs, and experimental duration; and it contained seven single-factor GCFs experiments (i.e., W, $eCO_2$, PPT +, PPT-, N deposition, P addition, and LUC), and only four combined-factors (i.e., W × $eCO_2$, N × PPT+, N × P and N × P × K additions) because of the limited number of multifactor studies (Supplementary Dataset 3). To explore the GCFs-induced effects specific for microbial groups and biomes based on the data availability, we binned the dataset by such microbial groups as fungi, bacteria, and the specialized microbes (denitrifier, 46 observations; nitrifier, 50 observations; diazotroph, 42 observations; P mineralizer, 17 observations; methanotroph, 21 observations; and methanogen, 2 observations), and by such biome types as agriculture, tundra, temperate/boreal forest, tropical/subtropical forest, Mediterranean vegetation, grassland, desert, and wetland.

**Calculation of the individual RRs.** The effect of GCFs on individual variable was estimated for each case study and calculated as the natural logarithm-transformed (ln) RR

$$RR = \ln\left(\frac{\overline{X_t}}{\overline{X_c}}\right) = \ln\left(\overline{X_t}\right) - \ln\left(\overline{X_c}\right), \quad (1)$$

where $\overline{X_t}$ and $\overline{X_c}$ are the means of the concerned variable in the treatment and control, respectively. However, the GCFs effect on soil pH was represented as change in soil pH, i.e., pH difference between treatment and control according to a previous meta-analysis[50]. Its variances ($v$) were calculated as:

$$v = \frac{S_t^2}{n_t \overline{X_t}^2} + \frac{S_c^2}{n_c \overline{X_c}^2} \quad (2)$$

where $n_t$ and $n_c$ are the sample sizes of the variable in the treatment and control, respectively; $s_t$ and $s_c$ are the standard deviations of the variable in the treatment and control, respectively.

**Microbial alpha diversity.** Richness and Shannon index are highly recommended when analyzing microbial alpha diversity[11–14,20,26]. Microbial richness metric is frequently reported by OTU, Chao, and ACE in the literature, while different studies use different metrics (Supplementary Dataset 1). The difference in RRs among OUT, Chao, and ACE within each case study was tested by a fixed-effect model with the moderator of metric of richness using the R package of *metafor*[51]. We chose the fixed-effect model because of the technical problems to conduct a random-effect model by few data points[51]. A total of 434 case studies reported at least two types of alpha diversity metrics, of which only 20 (<5%) showed sig-nificant differences (Supplementary Fig. 11) and different metrics had very little effect on the RR of microbial richness to GCFs (Supplementary Fig. 12a), indicating that using different metrics would not introduce much bias in richness analysis. Therefore, we chose a random-effect model to calculate the overall RR of richness for each case study.

**Beta diversity and community structure.** The ordination analysis is a key method for analyzing community of ecological data, e.g., principal component analysis, redundancy analysis, correspondence analysis, principal coordinate analysis, and nonmetric multidimensional scaling, and so on[52]. These techniques identify the similarity between species or samples generally by projecting them onto two dimensions in such a way that similar species or samples are clustering, while dissimilar ones fall apart[52]. In other words, these ordination plots display beta diversities within each treatment and the community structure differences among treatments[52,53]. However, as the meta-analysis is based on one-dimensional data, we used the following method to conduct the meta-analysis using the community data from the ordination plots with two dimensions. In specific, the effect of global change on community structure is considered if the distance between control and treatment is significantly greater than the distance within group, i.e., the positions of samples for control and treatment are not overlapped. The effect of global change on beta diversity is considered if the distance within treatment is sig-nificantly different from that within control (Supplementary Fig. 13).

Therefore, we firstly extracted the positions of samples on first two ordination axes (Supplementary Dataset 2). Second, Euclidean distances among different samples were calculated with the R packages of vegan[54], including the distance within control ($D_c$), that within treatment ($D_t$), and that between control and treatment ($D_b$). Third, we calculated the means, standard deviations, and sample sizes of $D_c$, $D_t$, $D_b$, and overall $D_c$ and $D_c$ ($D_c + D_c$), respectively. Finally, the RR of

microbial community structure (RR$_{Structure}$) and RR of beta diversity (RR$_{Beta}$) were calculated as:

$$RR_{Structure} = \ln\left(\frac{\overline{D_b}}{\overline{D_c + D_t}}\right), \qquad (3)$$

$$RR_{Beta} = \ln\left(\frac{\overline{D_t}}{\overline{D_c}}\right), \qquad (4)$$

where $\overline{D_c}$, $\overline{D_t}$, $\overline{D_b}$ and $\overline{D_c + D_t}$ are the means of the $D_c$, $D_t$, $D_b$ and $D_c + D_c$, respectively (Supplementary Fig. 13). Their corresponding variances were calculated as Eq. (2).

RR$_{Structure}$ < 0 indicates that the GCF has no effect on microbial community structure; and a greater positive value of RR$_{Structure}$ indicates a greater magnitude of change in the community structure. If RR$_{Beta}$ < 0, beta diversity was considered to be decreased by GCF, otherwise to be increased by GCF (Supplementary Fig. 13). Ordination methods inconsistently influence the responses of microbial beta diversity and community structure to GCFs, and most of the GCFs effects were not significant based on the omnibus test (Supplementary Fig. 12b, c). These results suggest that the current calculation makes it possible to compare the responses of microbial beta diversity and community structure to GCFs from different studies.

**Microbial ecosystem functionality**. We recoded 16 microbial functions related to soil biogeochemical cycling from the papers or their cited papers or the papers from the same experiment, including (1) microbial respiration, (2) net N mineralization rate, (3) nitrification activity, (4) denitrification activity, (5) biological N$_2$ fixation; and 11 types of enzymes related to soil C, N, and P cycling. They included oxidative C-cycling enzymes: (6) phenol oxidase (oxidize phenols using oxygen) and (7) peroxidase (oxidize aromatic and aliphatic hydrocarbons using peroxide); hydrolytic C-cycling enzymes: (8) α-1,4-glucosidase (starch degradation), (9) β-1,4-glucosidase (hydrolyze glucose from cellobiose and cellulose oligomers), (10) cellobiohydrolase (cellulose degradation), (11) β-1,4-xylosidase (hydrolyze xylose from hemicellulose and extracellular polysaccharides), and (12) invertase (hydrolysis of sucrose to glucose and fructose); N-cycling enzymes: (13) N-acetyl-β-glucosaminidase (chitin and peptidoglycan degradation), (14) L-leucine aminopeptidase (hydrolyses leucine and other hydrophobic amino acids from the N terminus of polypeptides), and (15) urease (catalyzes the hydrolysis of urea into ammonia and carbon dioxide); and P-cycling enzymes: (16) phosphatase (mineralize organic P into phosphate)[55–57]. Therefore, these indices are good proxies of processes driving soil biogeochemical cycling, and are frequently used to estimate the ecosystem multifunctionality of microbial communities[20,58]. The random-effect model was used to calculate RRs of oxidative C-cycling enzymes, hydrolytic C-cycling enzymes, N-cycling enzymes, and overall microbial ecosystem functions (all of the 16 functions) for each observation. We also analyzed the relationships between RR of alpha diversity and RR of functionality for specialized microbes except for methanogenic and methanotrophic communities because the limited data of functions (2 points for methane emission rate and 3 points for methane oxidation rate, respectively; Supplementary Dataset 1).

**Calculation of the overall RR**. The mixed-effect model was used to calculate the overall RR and corresponding 95% confidence intervals of target variables to different GCFs with a moderator of GCF types. It was also used to compare the RRs of target variables to each GCF among different microbial groups and biome types by the omnibus test ($Q_M$). The groups with small sample size (<5) were removed in these analyses. If the 95% confidence intervals for one RR overlapped with zero, then it was considered as an insignificant response to GCF.

**Model selection**. Model selection was based on AIC corrected (AIC corrected for small samples). The relative importance value for a particular predictor was equal to the sum of the Akaike weights (probability that a model is the most plausible model) for the models in which the predictor appears. Hence, a predictor that is included in models with large Akaike weights will receive a high importance value. These values can be regarded as the overall support for each variable across all models. A cutoff of 0.8 is set to differentiate between important and nonessential predictors. For this purpose, we used the R packages of gmulti[59].

Six types of candidate predictors were considered in the model selection analysis, that are, (1) climate factors, including MAT and MAP; (2) GCFs regimes, including GCF types, magnitude of W, percentage changes in precipitation, LUC types, N addition rate, P addition rate, K addition rate, and study duration; (3) changes in soil resources conditions, including the RRs of soil C, N, and C:N; (4) changes in soil pH; (5) microbial groups; and (6) biome types. For microbial community structure, the absolute values of RRs of soil C, N, C:N, and pH change were used in the model selection analysis because it is meaningless to consider community structure increasing or decreasing, we consequently hypothesis that greater absolute change in soil C, N, C:N, and pH would result in greater change in community structure (see the calculation of community structure). We did not

conduct this analysis for eCO$_2$ and W × eCO$_2$ due to the limited sample sizes (Supplementary Dataset 1 and 3).

**Reporting summary**. Further information on research design is available in the Nature Research Reporting Summary linked to this article.

## Data availability
The authors declare that the data supporting the findings of this study are available in Supplementary Dataset 1 and 2. Source data are provided with this paper.

## Code availability
The authors declare that the R (R-3.6.2) codes used to generate the results and figures reported in this study are available in Supplementary Software 1.

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

## Acknowledgements

We thank all the researchers whose data were used in this global synthesis. We appreciate Alexa McKay for their valuable comments. We also thank Noah Fierer and Ben Bond-Lamberty for the suggestions on this study. This work was financially supported by the National Natural Science Foundation of China (31901293), the Young Elite Scientists Sponsorship Program by China Association for Science and Technology (2018QNRC001), the National Key Technology Research and Development Program of China (No. 2011BAD37B01), the Program for Changjiang Scholars and Innovative Research Team in University (IRT_15R09), and the Heilongjiang Touyan Innovation Team Program for Forest Ecology and Conservation.

## Author contributions

Z.Z. and C.W. conceived the study, and developed it with Y.L.; Z.Z. collected and organized the data, and wrote the first draft of the paper; all authors contributed to discussing the results, writing, and editing the paper.

## Competing interests

The authors declare no competing interests.
