## [Peer Review File · Nature Communications]

Reviewers' Comments:

Reviewer #1:

Remarks to the Author:

Using the meta-analysis technique, the manuscript presents the results of how global changes affect microbial diversity and functionality in terrestrial ecosystems. Their main results are: 1) global changes do not always lead to a reduction in microbial diversity as reported for aboveground communities; 2) global-change-induced shifts in microbial α diversity can be predominately explained by the changed soil pH; 3) global change-induced changes in microbial diversity do not mirror the changes in microbial biomass production and functionality. This study produces useful results for understanding the effect of global changes on microbial diversity and functionality. Although the findings in this study have some implications for policy-making for biodiversity conservation under global changes, I have some significant concerns on the conclusion. Below, I listed some major points that should be concerned and addressed.

1. In Line 99, Line 102 and Line 105, the authors used the expression "a net increase in microbial diversity" and "net loss of microbial diversity". I think the use of "net increase or loss" is inappropriate because of two reasons. First, the numbers of observations measuring microbial α diversity and β diversity are not equal, or the numbers are equal but they (α diversity and β diversity) come from different case studies. Second, the microbial diversity is not equal to α diversity plus β diversity, especially the data were presented as a response ratio (logarithmic form). To address these problems, I suggest the authors to use paired α diversity and β diversity from the same case studies to calculate microbial (total) diversity and improve calculation method to combine these two metrics of diversity.

2. Why did the authors use the RR (response ratio) of pH to indicate the changes of pH under global changes? The pH is calculated as the negative value of logarithmic H^+ concentration. Therefore, the response ratio of pH is inappropriate and meaningless. I suggest the authors use pH unit change (i.e., $pH(\text{treatment}) - pH(\text{control})$) to indicate the effects of global change factors on soil pH (see Tian and Niu, 2015 ERL and Meng et al. 2019 ERL).

3. This study included 7 single-factor global changes, but the combined-factor only included four (i.e., $W \times eCO_2$, $N \times PPT+$, $N \times P$, and $N \times P \times K$ addition). The authors should clarify why they only included these four combined-factor experiments.

4. In this study, 14 microbial functions related to soil C and nutrient cycling were included. However, the key function "microbial respiration or decomposition" (measured in the field or in lab-incubation) was missed here. Microbial respiration is a key function related to soil C cycling, plays a key role in biogeochemistry, and would be more direct to indicate microbial function of C cycling than enzymes. Moreover, it seems that the authors combined these 14 functions to get an overall response ratio within each global change (Fig. 3a). The authors should clarify how they calculate the key variable "microbial ecosystem functionality".

5. In Fig. 2 and Fig. 3, the relationships were obtained based on the overall RR of pH, the overall RR of microbial α diversity, and the overall RR of functionality within each global change (11 individual points). That is, these relationships are across global change factors rather than within each global change factor. In addition, in Line 141, the authors declared that soil pH is the predominate predictor of the microbial α diversity. As we know, the N addition often caused soil acidification (also the results under land-use change and $N \times P \times K$ addition presented in Extended Data Fig. 5). It is not surprising to understand the role of changes of pH in regulating microbial α diversity under N enrichment (also land-use change and $N \times P \times K$ addition as presented in Extended Data Fig. 2). However, under other global change, the mechanism of how changes of pH regulate microbial α diversity remains unclear

and is not well addressed. The relationships within each global change factor are also very important, and can help to verify the universality of these relationships across global change factors. Given these reasons mentioned, I am not convinced about this conclusion in Line 141-142. I suggest the authors show and discuss these results within each global change factor in their manuscript.

Specific comments:

Line 95-96. As one of the objectives of this study is to investigate "Are the effects similar to those for plants and animals?". I suggest that the authors should cite and present the previous results about the effects of global changes on diversity of plants and animals.

Line 123-124. This sentence did not express the meanings of these two figures (Fig. 2b and Extended Data Fig. 4). What did you mean by using "co-stimulated"?

Line 310-311. The authors should list the keywords and their combinations used to search appropriate papers.

Line 323-324. The authors have listed plantation, pasture, or agricultural land. Here, what did "secondary ecosystem" refer to?

Lin 345. Why did you choose the fixed-effect model?

Line 362 and Line 365. was considered what? I guess something was missed here.

Line 403. of the effect of different global changes on target variables

Line 501-505. In Fig. 2 and Extended Data Fig. 2, what did the soil C, soil N and pH refers to, RR of these variables or the basic value of these variable in the control plot? Please clarify.

Line 507-510. The authors should clarify the calculation of "model coefficients" as presented in these figures. As shown in these figures, the variations of the coefficients in each panel is very large. Have you standardized these coefficients or showed the original data? As we know, different moderators are with different range and unit, for example, MAT and MAP in your figures.

Reviewer #2:

Remarks to the Author:

This manuscript assesses the impact of global changes on soil microbial diversity and functionality. The meta-analysis is based on a total of 913 experimental studies performed in the field. I like the concept of this work and think this is an important contribution. However, in my view further data-analysis is need to further substantiate the conclusions. Many variables and studies are lumped together. For instance, it is unclear whether the review is based on bacterial diversity or also on fungal diversity. Bacteria and fungi are fundamentally different and I would not lump them together. At least the data analysis need to be also performed separately for each group si that it can be tested whether similar conclusions can be obtained for each of the microbial groups. The location of the different experiments needs to be shown on a map (figure for the supplement). Are these studies mainly performed in temperate zones or also in the tropics or in arctic regions? This will be visible from such a map and I wonder whether location affects the outcome of the result (e.g. it is not surprising to find a positive effect of warming on microbial diversity in a temperate or arctic location – effects in the tropics may be different). Furthermore, different land use types (grassland, forest, agricultural arable land)

are included in the analysis (and these are again all lumped together). It is necessary to at least test whether the effects are dependent on habitat/land use type (or the habitats with the most data points). I understand that lumping of the data will enhance strength, but I am concerned that the conclusions could be different if specific land use types (forest, grassland or agricultural land) or microbial groups (fungi or bacteria) are analysed separately.

Specific points:

It would be useful to provide a table summarizing the number of studies per global change factor (separate column for microbial diversity measures and microbial functionality), microbial group and land use type.

Figure 2 is appealing and gives a nice mechanistic explanation of how global change factors could affect microbial richness.

Methods: high throughput studies were used for this analysis. What type of high throughput (Illumina, 454, ??). Please specify

Reviewer #3:

Remarks to the Author:

The Manuscript " Impacts o global changes on soils microbial diversity and functionality" by Zhou, Wang and Luo used meta-analysis approach to reports that global changes do not always lead to reduction in microbial diversity. Furthermore changes in microbial diversity does not results in altered soil functionality. This is an important topic and authors should be commended for the effort to put together these data. However, I have a number of major technical and analytical concerns and at this moment conclusions seems not be supported.

1. First, microbial diversity studies have made a significant advancements in terms of both drivers and linkage with ecosystem functionings. Vast majority of studies 2015 (studies which used next-generation sequencing and appropriate analytical approaches) found consistent positive relationship between soil microbial diversity and ecosystem functions (e.g. Jing et al., 2016; Delgado-Baquerizo et al 2016; Nature Communications). Suggesting that we have very little idea on microbial BEF is not correct anymore. Saying that, new and more balanced studies are definitely needed.

2. Not sure- how data is distributed- this is most important context that needs to be explicitly considered. If most of studies come from temperate and high carbon soils, then loss of carbon or agriculture practices (that will increase soil pH) is expected to increase diversity. It would be nice to know how many studies was based in drylands (43% global land mass) and low organic C systems - and how these samples performed under increased temperature or drought conditions.

3. Functional measures (particularly enzyme assay) is incredibly susceptible to vary based on how sampling was carried out and between team and laboratories. I suspect most of differences authors picking up could be explained by this artefacts in data. Such data in my opinion should always be generated in one lab following the same protocols- that is definitely not the case here. At least authors can re-run a certain percentage of samples to validate these findings.

4. Not all functions are carried our by significant number of species- there is growing evidence that some key soil functions (methane cycle, nitrification, denitrifcantion, pollutant degradations) that are carried out by restricted number of phylogenetic groups - are extremely vulnerable to loss in diversity. It is possible that bias in data and focusing of proxy of functional measures, i.e. enzyme activities

(which is dominant this dates) rather than real functions have biased conclusions?

6. Overall, I believe, authors need some more data to balance the dataset, use range of real functions (both general and specialised) and not just proxy, and generate some actual data to validate this result. In my opinion, in the current form this manuscript does not advance science.

Reviewers' comments:

Reviewer #1 (Remarks to the Author):

Using the meta-analysis technique, the manuscript presents the results of how global changes affect microbial diversity and functionality in terrestrial ecosystems. Their main results are: 1) global changes do not always lead to a reduction in microbial diversity as reported for aboveground communities; 2) global-change-induced shifts in microbial α diversity can be predominately explained by the changed soil pH; 3) global change-induced changes in microbial diversity do not mirror the changes in microbial biomass production and functionality. This study produces useful results for understanding the effect of global changes on microbial diversity and functionality. Although the findings in this study have some implications for policy-making for biodiversity conservation under global changes, I have some significant concerns on the conclusion. Below, I listed some major points that should be concerned and addressed.

1. In Line 99, Line 102 and Line 105, the authors used the expression “a net increase in microbial diversity” and “net loss of microbial diversity”. I think the use of “net increase or loss” is inappropriate because of two reasons. First, the numbers of observations measuring microbial α diversity and β diversity are not equal, or the numbers are equal but they (α diversity and β diversity) come from different case studies. Second, the microbial diversity is not equal to α diversity plus β diversity, especially the data were presented as a response ratio (logarithmic form). To address these problems, I suggest the authors to use paired α diversity and β diversity from the same case studies to calculate microbial (total) diversity and improve calculation method to combine these two metrics of diversity.

Response: Thanks for your comments. As you suggested, we selected the data that reported both microbial alpha diversity and beta diversity synchronously and did paired analyses, of which the results (**Supplementary Fig. 3**) were very consistent with the results presented in **Fig. 1** in the main body. In essence, it is technically problematic to calculate the total diversity by combining alpha diversity and beta diversity, because these metrics have different definitions and perspectives, and are calculated by different formulas. We do not think it is a big problem to conclude with a positive effect of GCs on the diversity if GCs have a significant positive effect on one metric but an insignificant (or neutral) effect on the other. To be more accurate, we revised the expressions of “a net increase in microbial diversity” and “net loss of microbial diversity” in the text.

2. Why did the authors use the RR (response ratio) of pH to indicate the changes of pH under global changes? The pH is calculated as the negative value of logarithmic H^+ concentration. Therefore, the response ratio of pH is inappropriate and meaningless. I suggest the authors use pH unit change (i.e., $pH(\text{treatment}) - pH(\text{control})$) to indicate the effects of global change factors on soil pH (see Tian and Niu, 2015 ERL and Meng et al. 2019 ERL).

Response: Thanks. Following your suggestion, we used change in soil pH directly, instead of RR of pH, throughout the revised version; all of the original patterns are maintained. Actually, RR of pH and change in pH are highly correlated (see the figure below).

Relationships between response ratios (RR) of soil pH (ln(treat/control)) and change in soil pH (treat-control).

3. This study included 7 single-factor global changes, but the combined-factor only included four (i.e., W×eCO₂, N×PPT+, N×P, and N×P×K addition). The authors should clarify why they only included these four combined-factor experiments.

Response: The dataset only includes four combined-factors because of the limited sample sizes for other combinations. Clarified in the revised method.

4. In this study, 14 microbial functions related to soil C and nutrient cycling were included. However, the key function “microbial respiration or decomposition” (measured in the field or in lab-incubation) was missed here. Microbial respiration is a key function related to soil C cycling, plays a key role in biogeochemistry, and would be more direct to indicate microbial function of C cycling than enzymes. Moreover, it seems that the authors combined these 14 functions to get an overall response ratio within each global change (Fig. 3a). The authors should clarify how they calculate the key variable “microbial ecosystem functionality”.

Response: We agree with you, added microbial respiration in the dataset, and clarified the calculation of overall microbial functionality in the method. We established the correlations between alpha diversity and different functions (i.e., microbial respiration, N mineralization, oxidative C-cycling enzymes, hydrolytic C-cycling enzymes, N-cycling enzymes, and P-cycling enzymes), and the conclusions are maintained. We also analyzed the diversity–functionality relationships by several specialized microbial groups based on the data availability. Please also refer to our response to **Reviewer #3’s comments** for more revisions associated with the functionality.

5. In Fig. 2 and Fig. 3, the relationships were obtained based on the overall RR of pH, the overall RR of microbial α diversity, and the overall RR of functionality within each global change (11 individual points). That is, these relationships are across global change factors rather than within each global change factor. In addition, in Line 141, the authors declared that soil pH is the predominate predictor of the microbial α diversity. As we know, the N addition often caused soil acidification (also the results under land-use change and N×P×K addition presented in Extended Data Fig. 5). It is not surprising to understand the role of changes of pH in regulating microbial α diversity under N enrichment (also land-use change and N×P×K addition as presented in Extended Data Fig. 2). However, under other global change, the mechanism of how changes of pH regulate microbial α diversity remains unclear and is not well addressed. The relationships within each global change factor are also very important, and can help to verify the universality of these relationships across global change factors. Given these reasons mentioned, I am not convinced about this conclusion in Line

141-142. I suggest the authors show and discuss these results within each global change factor in their manuscript.

Response: Thanks for your suggestion. We updated the dataset until February 2020, which is increased by 35%. We conducted the model selection to explore the relative importance for the candidate predictors within each GC factor (except for eCO₂ and W×eCO₂), and added the associated results and discussion in the revision (**Fig. 2-4, and Supplementary Fig. 4-6**). The limited sample prohibited such analysis for eCO₂, but the correlation analysis showed that RRs of alpha diversity to eCO₂ were decoupled with all of the candidate predictors (see the table below).

Correlation coefficients between RR of richness and Shannon index to eCO₂ and candidate predictors.

Predictor	RR of Richness			RR of Shannon index		
	R	p	N	r	p	N
MAT	-0.18	0.28	39	0.31	0.15	22
MAP	-0.31	0.06	39	0.09	0.71	22
Magnitude	-0.09	0.57	39	-0.39	0.07	22
Duration	0.20	0.21	39	0.12	0.59	22
Change in pH	0.08	0.71	24	-0.30	0.34	12
RR of soil C	-0.05	0.81	23	0.18	0.67	8
RR of N	-0.10	0.68	20	0.49	0.26	7
RR of CN	0.25	0.28	20	-0.29	0.53	7

Specific comments:

Line 95-96. As one of the objectives of this study is to investigate “Are the effects similar to those for plants and animals?”. I suggest that the authors should cite and present the previous results about the effects of global changes on diversity of plants and animals.

Response: Thanks. Revised.

Line 123-124. This sentence did not express the meanings of these two figures (Fig. 2b and Extended Data Fig. 4). What did you mean by using “co-stimulated”?

Response: Revised as: RRs of microbial richness ($R^2 = 0.87$, $P = 0.004$) and Shannon index ($R^2 = 0.84$, $P = 0.004$) significantly increase as the changes in soil pH increased (Fig. 2b).

Line 310-311. The authors should list the keywords and their combinations used to search appropriate papers.

Response: Added in the method.

Line 323-324. The authors have listed plantation, pasture, or agricultural land. Here, what did “secondary ecosystem” refer to?

Response: Secondary ecosystem is defined as naturally-developed stand with native species from the harvest stand, pasture, agricultural land, and other disturbed stands; and it differs from plantation and pasture mainly in terms of human activity involved in the stand establishment and development (Don et al., 2011; Zhou et al., 2018). We added a note in the text.

Don, A., Schumacher, J. & Freibauer, A. Impact of tropical land-use change on soil organic carbon stocks—a meta-analysis. *Global Change Biol.* **17**, 1658–1670 (2011).

Zhou, Z., Wang, C. & Luo Y. Effects of forest degradation on microbial communities and soil carbon cycling: A global meta-analysis. *Global Ecol Biogeogr.* **27**, 110–124 (2018).

Lin 345. Why did you choose the fixed-effect model?

Response: Ideally, the random-effect model should be better due to the heterogeneity of different studies, but technically it is impossible to conduct a random-effect model by two or three data points (the paired richness index among OTU, Chao, and ACE) in which the parameters to be estimated outnumber the observations. Therefore, we used the fixed-effect model to compare the effects of different metrics on the RRs of richness, but chosen a random-effect model to calculate the overall RR of richness for each case study. We clarified it in the revised method.

Line 362 and Line 365. was considered what? I guess something was missed here.

Response: Sorry for the confusion. We clarified and revised as “The ordination analysis is a key method for analyzing community of microbial data, e.g., principal component analysis (PCA), redundancy analysis (RDA), correspondence analysis (CA), principal coordinate analysis (PCoA), and non-metric multidimensional scaling (NMDS), so on (Paliy & Shankar, 2016). These techniques identify the similarity between species or samples generally by projecting them onto two dimensions in such a way that similar species or samples are clustering, while dissimilar ones fall apart (Paliy & Shankar, 2016). In other words, these ordination plots display beta diversities within each treatment and the community composition differences among treatments (Cajo & Braak, 1983; Paliy & Shankar, 2016). However, as the meta-analysis is based on one-dimensional data, we used the following method to conduct the meta-analysis using the community data from the ordination plots with two dimensions. In specific, the effect of global change on community composition is considered if the distance between control and treatment is significantly greater than the distance within group, i.e., the positions of samples for control and treatment are not overlapped. The effect of global change on beta diversity is considered if the distance within treatment is significantly different from that within control (**Supplementary Fig. 10**).”

Cajo, J. F. & Braak, T. Principal components biplots and alpha and beta diversity. *Ecology* **64**, 454–462 (1983).
Paliy, O. & Shankar, V. Application of multivariate statistical techniques in microbial ecology. *Mol. Ecol.* **25**, 1032–1057 (2016).

Line 403. of the effect of different global changes on target variables

Response: Revised.

Line 501-505. In Fig. 2 and Extended Data Fig. 2, what did the soil C, soil N and pH refers to, RR of these variables or the basic value of these variable in the control plot? Please clarify.

Response: All of these soil variables are the corresponding RRs. Clarified.

Line 507-510. The authors should clarify the calculation of “model coefficients” as presented in these figures. As shown in these figures, the variations of the coefficients in each panel is very large. Have you standardized these coefficients or showed the original data? As we know, different moderators are with different range and unit, for example, MAT and MAP in your figures.

Response: It is the original model coefficients without standardization, which were transferred by the fourth root for better visualization in the revised manuscript. Clarified (**Supplementary Fig. 5**).

=====

Reviewer #2 (Remarks to the Author):

This manuscript assesses the impact of global changes on soil microbial diversity and functionality. The meta-analysis is based on a total of 913 experimental studies performed in the field. I like the concept of this work and think this is an important contribution. However, in my view further data-analysis is need to further substantiate the conclusions. Many variables and studies are lumped together. For instance, it is unclear whether the review is based on bacterial diversity or also on fungal diversity. Bacteria and fungi are fundamentally different and I would not lump them together. At least the data analysis need to be also performed separately for each group si that it can be tested whether similar conclusions can be obtained for each of the microbial groups. The location of the different experiments needs to be shown on a map (figure for the supplement). Are these studies mainly performed in temperate zones or also in the tropics or in arctic regions? This will be visible from such a map and I wonder whether location affects the outcome of the result (e.g. it is not surprising to find a positive effect of warming on microbial diversity in a temperate or arctic location – effects in the tropics may be different). Furthermore, different land use types (grassland, forest, agricultural arable land) are included in the analysis (and these are again all lumped together). It is necessary to at least test whether the effects are dependent on habitat/land use type (or the habitats with the most data points). I understand that lumping of the data will enhance strength, but I am concerned that the conclusions could be different if specific land use types (forest, grassland or agricultural land) or microbial groups (fungi or bacteria) are analysed separately.

Response: Thanks for your comments. According to your suggestion, we did the following revisions: (1) We added a map (**Supplementary Fig. 1**) showing the distribution of sampling sites in this study, and summarized the basic characteristics, such as ranges of climates, experimental regimes, etc. in the **Supplementary Dataset 3**. (2) We binned the data and re-analyzed for the microbial responses to GCs by microbial groups [i.e., fungi, bacteria, and specialized microbes (denitrifier, 46 observations; nitrifier, 50 observations; diazotroph, 42 observations; P mineralizer, 17 observations; methanotroph, 21 observations; and methanogen, 2 observations)] and biome types (i.e., agriculture, tundra, boreal/temperate forest, tropical/subtropical forest, Mediterranean vegetation, grassland, desert, and wetland) (see **Fig. 2-4, and Supplementary Fig. 8**). Microbial groups and biome types were also added in the model selection analysis (**Fig. 2, Supplementary Fig. 4-6**). (3) All the results mentioned microbial groups, biomes, and climates were discussed in the main body.

Specific points:

It would be useful to provide a table summarizing the number of studies per global change factor (separate column for microbial diversity measures and microbial functionality), microbial group and land use type.

Response: Thanks for your suggestion. Added (**Supplementary Dataset 3**).

Figure 2 is appealing and gives a nice mechanistic explanation of how global change factors could affect microbial richness.

Response: Thanks.

Methods: high throughput studies were used for this analysis. What type of high throughput (Illumina, 454, ??). Please specify

Response: The high throughput data compiled in the dataset included Illumina, 454, and ABI. Clarified and specified in the method.

=====

Reviewer #3 (Remarks to the Author):

The Manuscript "Impacts of global changes on soils microbial diversity and functionality" by Zhou, Wang and Luo used meta-analysis approach to reports that global changes do not always lead to reduction in microbial diversity. Furthermore changes in microbial diversity does not results in altered soil functionality. This is an important topic and authors should be commended for the effort to put together these data. However, I have a number of major technical and analytical concerns and at this moment conclusions seems not be supported.

1. First, microbial diversity studies have made a significant advancements in terms of both drivers and linkage with ecosystem functionings. Vast majority of studies 2015 (studies which used next-generation sequencing and appropriate analytical approaches) found consistent positive relationship between soil microbial diversity and ecosystem functions (e.g. Jing et al., 2016; Delgado-Baquerizo et al 2016; Nature Communications). Suggesting that we have very little idea on microbial BEF is not correct anymore. Saying that, new and more balanced studies are definitely needed.

Response: Thanks for your comments. We updated the dataset until February 2020 and tried our best to provide a balanced synthesis. The dataset is increased by 35%, which included several specialized groups of microbes (denitrifier, nitrifier, diazotroph, P mineralizer, methanotroph, and methanogen) and corresponding functions. We re-analyzed the data and found significant negative diversity–functionality relationships for denitrifier and nitrifier and decoupled correlations for diazotrophic and P mineralizer (**Fig. 6**). The overall conclusion that GC-induced changes in microbial diversity are not necessary to mirror the shifts in functionality is maintained even for the specialized microbial groups.

2. Not sure- how data is distributed- this is most important context that needs to be explicitly considered. If most of studies come from temperate and high carbon soils, then loss of carbon or agriculture practices (that will increase soil pH) is expected to increase diversity. It would be nice to know how many studies was based in drylands (43% global land mass) and low organic C systems - and how these samples performed under increased temperature or drought conditions.

Response: A good idea. We provided a map showing the distribution of sampling points (**Supplementary Fig. 1**). Also, we re-analyzed the dataset by biomes (i.e., agriculture, tundra, boreal/temperate forest, tropical/subtropical forest, Mediterranean vegetation, grassland, desert, and wetland), and examined the effects of MAT and mean annual precipitation (MAP) on the RRs of richness and Shannon index within each global change factor. All these results by biomes and climates were added and then discussed in the main body. More details please refer to our responses to **Reviewer #2's general comments**.

3. Functional measures (particularly enzyme assay) is incredibly susceptible to vary based on how sampling was carried out and between team and laboratories. I suspect most of differences authors picking up could be explained by this artefacts in data. Such data in my opinion should always be generated in one lab following the same protocols- that is definitely not the case here. At least authors can re-run a certain percentage of samples to validate these findings.

Response: Non-standardized methods in the literature for enzyme assay may result in uncertainty in global meta-analyses (Dick, 2011; Nannipieri et al., 2018), such as fresh vs. air dried soils, difference in incubation temperature and time, whether to add sodium hydroxide to stop the reaction, difference in storage (both time and temperature) of the soils before assay, just to name a few (Dick, 2011; Nannipieri et al., 2018). In reality, however, such information is not clearly reported in all publications. For example, 116 of 270 observations in the dataset did not report the incubation temperature for enzymes clearly (**Supplementary Datasets 1**);

few studies reported the storage time of the soil before assay. To minimize the heterogeneity and draw useful signals, meta-analysis is one of suitable approaches. First, the calculation of RR (i.e., $\ln(\text{treat/control})$) to standardize the raw data can partly eliminate the potential heterogeneity from enzyme assay, sampling, etc. Second, the random-effect model used for calculating the RRs of microbial functions to global changes in the revised version treated the heterogeneity of methods and sample characteristics as purely random (Viechtbauer, 2010). Overall, it is technically impossible to take the details of enzyme assay methods for large-scale synthesis.

To do our best, we re-analyzed the correlations between alpha diversity and specific functions based on the data availability (i.e., microbial respiration, N mineralization, oxidative C-cycling enzymes, hydrolytic C-cycling enzymes, N-cycling enzymes, and P-cycling enzymes). We found non-significant positive effect of diversity on functions (**Fig. 4**). With consistent methods, we detected significant negative relationships between RRs of richness and N mineralization rate (**Supplementary Fig. 9**) and significant negative diversity–functionality relationships for denitrifier and nitrifier (**Fig. 6**). Therefore, the conclusion that GC-induced changes in microbial diversity is not necessary to mirror the shifts in functionality should be valid despite the variability induced by enzyme assay in different studies.

Dick, W. A. Development of a soil enzyme reaction assay. In *Methods of Soil Enzymology* **9**, 71-84 (2011).
Nannipieri, P., Trasar-Cepeda, C. & Dick, R. P. Soil enzyme activity: a brief history and biochemistry as a basis for appropriate interpretations and meta-analysis. *Biol. Fert. Soils* **54**, 11-19 (2018).
Viechtbauer, W. Conducting meta-analyses in R with the metafor package. *J. Stat. Softw.* **36**, 1-48 (2010).

4. Not all functions are carried out by significant number of species- there is growing evidence that some key soil functions (methane cycle, nitrification, denitrification, pollutant degradations) that are carried out by restricted number of phylogenetic groups - are extremely vulnerable to loss in diversity. It is possible that bias in data and focusing of proxy of functional measures, i.e. enzyme activities (which is dominant this dates) rather than real functions have biased conclusions? Overall, I believe, authors need some more data to balance the dataset, use range of real functions (both general and specialised) and not just proxy, and generate some actual data to validate this result. In my opinion, in the current form this manuscript does not advance science.

Response: Thanks for your suggestion. We updated the dataset and tried our best to provide a balanced synthesis especially by considering specialized microbial group, biome type, and climate as suggested, and the original conclusions are maintained. Please refer to our responses to the other comments.

We hope that you find our revision satisfactory. Thank you very much!

Reviewers' Comments:

Reviewer #1:

Remarks to the Author:

I reviewed the previous version of this manuscript. Overall, I think the massive dataset assembled by the authors contributes to our understanding of the responses of soil microbial communities to global change factors. However, I still have a few serious concerns for the current version.

1. The common way of testing the diversity-functioning relationship is from either observational studies at many sites regionally or globally (but using the same method to quantify both diversity and functions) or experimental studies at single site (but manipulating richness or diversity, even for soil microbes). In both ways, the measurement method should be consistent and comparable among sites or treatments. The rationale in this study is to correlate responses of microbial diversity against responses of soil functions (e.g. C and N mineralization rates, enzyme activities) to global changes. Other factors could confound this correlation, such as changes in plant communities (diversity and productivity) and variations in the methods to quantify both microbial diversity and functioning among various studies. Therefore, I think the correlations reported (e.g. Fig. 5) are very uncertain and not strong enough to support the argument that "... advance the biodiversity-productivity-functionality relationships in microbial ecology..." (L253-254).

2. In soil microbial ecology, an increasingly accepted idea is that the physiologically narrow processes (e.g. nitrification) are controlled by microbial diversity, while the physiologically broad processes (e.g. decomposition) may not due to microbial redundancy. Therefore, different functions should be considered against microbial diversity differently. Moreover, microbial biomass can be more important than microbial diversity in controlling function. The relative role of diversity vs. biomass on function should be separated.

3. One major technical problem is that different studies in the dataset used different methods to measure variables related to microbial diversity and functioning. Even the authors used the response ratios (rather than the original values) in this synthesis, this problem could still affect the robustness of the correlations among different studies. For example, the microbial beta diversity is calculated from the community ordination plots by different methods (PCA, RDA, CA, NMDS, PCoA, L313-315), which makes the metric of "beta diversity" hardly comparable among different studies. Also, the variables for the "functioning" were also measured by study-specific method. For example, how the incubations were conducted to measure microbial respiration and N mineralization, how the assays were conducted to estimate enzyme activities. This method issue should be considered.

4. Overall, I think the core results do not advance the science on microbial diversity and functioning to a significant level, given the limitations listed above.

Reviewer #2:

Remarks to the Author:

This manuscript assesses the impact of global changes on soil microbial diversity and functionality. I evaluated this manuscript again. I think the authors did a very thorough job (also including many more studies in their meta-analysis) and answered my queries properly. I think this is an important study. The presented links between microbial biomass and functionality are also appealing. I have mainly minor comments. I must admit that the abstract is not very specific (a bit broad and a bit "vague") and I feel it is necessary to integrate a few clear conclusions to attract a broader readership and get many more citations of this work (see specific comments for suggestions).

Specific comments.

Please make sure all abbreviations are explained in the text (e.g. line 85: "PPT", "W", "LUC", perhaps it is explained or partly explained but I missed it).

Line 105: "Additionally, a significant increase in alpha diversity during the conversions from highly-diverse natural ecosystems to homogeneous agricultural monocultures (Supplementary Fig. 2) implies that changes in microbial alpha diversity are also uncoupled with the shifts in plant alpha diversity."

I think this is a nice observation and this should be mentioned in the abstract.

Line 126: "suggesting that rare species are more sensitive to GCs than common species, in agreement study with a previous study (26)." However, that previous study is not a meta-analysis? Right?

I thought this is a nice observation and worth to be mentioned in the abstract.

Line 117: "We also found that GCs greatly change microbial community composition regardless of the effects of GCs on microbial diversity mentioned above".

Perhaps also a conclusion that could be suitable for the abstract.

This study found that richness is often not strongly responding to GC. However, microbial composition (and the occurrence of rare taxa) changed substantially). I think it is important to mention in the discussion that other microbial community traits such as microbial network structure and characteristics (connectance, keystone taxa, etc.) have also potential (or a better potential) to unravel effects of GC factors on microbial communities.

Figure 5e and 5f: the y-axis for 5e shows "absolute functionality" and the y-axis for 5f shows "functionality". Please explain the difference.

Supplement Figure 1: The distribution of the different sampling sites shows that most studies used for this work are from China, the US and some from Europe. From some locations (e.g. Africa, Northern Europe/North Asia, South America, there are hardly (or none) data entries. This reviewer is a European and I am sure that for some of the assessed global change factors, studies have been missed. For instance, I am sure there are more studies that assessed the impact of nitrogen addition or land use change on microbial communities. It is important to write a qualifier about this in the discussion. I think with such a broad study, it is hard to include all studies, so for me the current analysis is absolutely fine. But it is important to mention this because it could affect the conclusion and outcome of the study (if more studies had been included). Also, it is important to state clearly that the data are largely from China, the US and Europe and that there are large gaps on the worldwide map.

Note from the editor: The report has been modified with additional comments from Reviewer 2 on the new comments from Reviewer 1. The comments from Reviewer 2 are shown with asterisks.

Reviewer 1 comments:

I reviewed the previous version of this manuscript. Overall, I think the massive dataset assembled by

the authors contributes to our understanding of the responses of soil microbial communities to global change factors. However, I still have a few serious concerns for the current version.

1. The common way of testing the diversity-functioning relationship is from either observational studies at many sites regionally or globally (but using the same method to quantify both diversity and functions) or experimental studies at single site (but manipulating richness or diversity, even for soil microbes). In both ways, the measurement method should be consistent and comparable among sites or treatments. The rationale in this study is to correlate responses of microbial diversity against responses of soil functions (e.g. C and N mineralization rates, enzyme activities) to global changes. Other factors could confound this correlation, such as changes in plant communities (diversity and productivity) and variations in the methods to quantify both microbial diversity and functioning among various studies. Therefore, I think the correlations reported (e.g. Fig. 5) are very uncertain and not strong enough to support the argument that "... advance the biodiversity-productivity-functionality relationships in microbial ecology..." (L253-254).

REVIEWER 2 RESPONSE: Obviously, it would be better if the same methods had been used. However, the authors compared a control treatment with a GC treatment for each study. Subsequently it is possible to standardize the data and compare many different studies using response ratios. I think this is fine.

I think the reviewer is right about the last statement. It is very strong and not really necessary and they could remove it:

Something like: "Overall, our findings INDICATE THAT THE RESPONSE OF microbial communities TO GLOBAL CHANGES are fundamentally different from those for macro-communities."

2. In soil microbial ecology, an increasingly accepted idea is that the physiologically narrow processes (e.g. nitrification) are controlled by microbial diversity, while the physiologically broad processes (e.g. decomposition) may not due to microbial redundancy. Therefore, different functions should be considered against microbial diversity differently. Moreover, microbial biomass can be more important than microbial diversity in controlling function. The relative role of diversity vs. biomass on function should be separated.

REVIEWER 2 RESPONSE: This point could be addressed. I think the reviewer makes a good point here and it is. Probably good to further discuss this issue. I find it very interesting and important (see also my comments below).

3. One major technical problem is that different studies in the dataset used different methods to measure variables related to microbial diversity and functioning. Even the authors used the response ratios (rather than the original values) in this synthesis, this problem could still affect the robustness of the correlations among different studies. For example, the microbial beta diversity is calculated from the community ordination plots by different methods (PCA, RDA, CA, NMDS, PCoA, L313-315), which makes the metric of "beta diversity" hardly comparable among different studies. Also, the variables for the "functioning" were also measured by study-specific method. For example, how the incubations were conducted to measure microbial respiration and N mineralization, how the assays were conducted to estimate enzyme activities. This method issue should be considered.

REVIEWER 2 RESPONSE: I think it is correct to use the response ratio (RR) because this makes it possible to compare different studies. Note that the authors used the RR. Various studies (e.g. with biomass data) also use the logRR or lnRR because that can improve the normality of the data (please verify if this is the case for your data-set). Note that some authors used the sampling variances as proposed in Nakagawa et al. to account for sampling uncertainty in each observation (correcting for differences in sample size among studies and the corresponding difference in uncertainty and robustness of the result) – see also Knapp & van der Heijden 2018, Nature Communication, for references.

The RR also enables to automatically standardize across studies that used different methods, so I think the concerns of the reviewer are addressed by the author. I think the supplemental figure 10 shows that only in a few cases diversity measures gave different results in different studies, which clearly indicates that the approach is valid. It probably would be good if the authors clarify in the figure legend of this figure how they then could test whether there was a significant difference between measures for a particular study / observation (e.g. I assume this is because each study had various replications for the control and the GC treatment). I realize now, however, that such a supplemental figure is not shown for the different beta diversity matrixes used (e.g. PCA, RDA, CA, NMDS, PCoA,). This should be included as well and the reviewer correctly points to this (line 382 & 400 actually says it is presented in supplementary figure 10, but only microbial alpha diversity indexes are compared there – so this figure is missing!). Also, the methods used to calculate beta-diversity are more diverse and as such perhaps pooling them all is perhaps more critical compared to the alpha diversity indexes. Therefore, such a figure is necessary and such issues need to be discussed (can also be a discussion for a supplement or in the methods section).

Moreover, figure 5c suggests that alpha diversity is not important for functionality. It is important to clearly state (and discuss) that the change in alpha diversity is negatively correlated with functionality. This does not necessarily say that microbial alpha diversity is not important for functionality because the observations in this meta-analysis are based on changes within experiments, not a global correlation between microbial diversity and functionality (e.g. various studies show that microbial alpha diversity is positively linked to functionality – Wagg et al. 2014 (PNAS), 2019 (Nature Comm), Jing et al. 2015 (Nature Comm), Delgado-Baquerizo et al. 2020 (NEE)). This issue needs to be carefully discussed. I like the notes about active versus dormant diversity microbes (line 223) and the importance of microbial biomass/microbial compostion versus microbial diversity. I think the observations in the figure will lead to discussion and will for sure develop the science in this area. Obviously more work is needed here. As such I appreciate this.

4. Overall, I think the core results do not advance the science on microbial diversity and functioning to a significant level, given the limitations listed above.

REVIEWER 2 RESPONSE: I think this is one of the largest meta-analysis I have seen on this subject and as such I do think it advances the science. The reviewer mainly points to technical issues, which is a matter of debate (see my response above for point 3), the question of novelty/relevance is not criticised. I am not aware of a similar meta-analysis in the literature, but it is always possible I missed something as there is such a huge number of studies coming out.

Reviewer #3:

Remarks to the Author:

Again, I would like to commend the authors for extensive re-analysis and revisions in the manuscript. I think most of technical questions raised by the reviewers have been addressed. Inclusion of functional communities in the manuscript is really appreciated.

My main concern regarding framing and presentation of the paper remains.

1. Authors use alpha- diversity liberally to proxy of biodiversity. As authors have mentioned biodiversity includes multiple attributes including, alpha, beta, gamma (etc) diversity. It is important that authors consistently use the term alpha diversity (richness or Shannon diversity) throughout the manuscript to avoid confusion.

2. The main finding of this manuscript is that GCs impacts on soil functions are explained by 'microbial composition' (I guess author mean microbial community structure here?) and microbial biomass and not alpha microbial diversity. I believe this will be a good addition in literature if authors can structure manuscript accordingly and not get distracted by Microbial BEF (this manuscript does not address this) and how it is similar or different from plant ecology. Authors have highlighted these findings (please see below comments) in the main text but title, abstract and conclusion do not reflect these key findings.

3. Title can be more informative, e.g. " microbial community structure and biomass and not alpha-diversity explain effect of GCs on soil functions.

4. L66- 76. This section is irrelevant- the manuscript is not addressing microbial BEF but impacts of GC- on microbial community attributes (alpha diversity, community structure and biomass) and if these can explain change in soil functions. I therefore suggest to delete this section.

5. In my opinion, following are key findings are (1) GCs greatly change microbial community (L116-122) and it explains significant ($R^2 = 0.58$) variation in soil functions; (2) microbial biomass positively correlated with soil functions ($R^2 = 0.51$) (L227-229) and biomass of specialised community are strongly linked to rate of specialised functions (L240-243). Currently these important findings are hidden in the main text and in my opinion, these should be reflected strongly in title, abstract and conclusion sections.

REVIEWERS' COMMENTS:

Reviewer #1 (Remarks to the Author):

I reviewed the previous version of this manuscript. Overall, I think the massive dataset assembled by the authors contributes to our understanding of the responses of soil microbial communities to global change factors. However, I still have a few serious concerns for the current version.

1. The common way of testing the diversity-functioning relationship is from either observational studies at many sites regionally or globally (but using the same method to quantify both diversity and functions) or experimental studies at single site (but manipulating richness or diversity, even for soil microbes). In both ways, the measurement method should be consistent and comparable among sites or treatments. The rationale in this study is to correlate responses of microbial diversity against responses of soil functions (e.g. C and N mineralization rates, enzyme activities) to global changes. Other factors could confound this correlation, such as changes in plant communities (diversity and productivity) and variations in the methods to quantify both microbial diversity and functioning among various studies. Therefore, I think the correlations reported (e.g. Fig. 5) are very uncertain and not strong enough to support the argument that "... advance the

biodiversity-productivity-functionality relationships in microbial ecology..." (L253-254).

Response: Thanks. We agree that it would be ideal if the same method had been used, but this study explored the impacts of global change factors in each study of the global dataset. Therefore we tried our best to standardize the data and to compare the global results with the meta-analysis. Meta-analysis did a good job confirming trends reported in individual papers worldwide in despite of its limitations. In addition, we removed the statement of 'advance the biodiversity-functionality relationships in microbial ecology' in both abstract and conclusion sections as suggested by the editor and other reviewers.

2. In soil microbial ecology, an increasingly accepted idea is that the physiologically narrow processes (e.g. nitrification) are controlled by microbial diversity, while the physiologically broad processes (e.g. decomposition) may not due to microbial redundancy. Therefore, different functions should be considered against microbial diversity differently. Moreover, microbial biomass can be more important than microbial diversity in controlling function. The relative role of diversity vs. biomass on function should be separated.

Response: Thanks for your valuable comments. We had analyzed the diversity-function relationships for several specialized microbes and the corresponding narrow function, and our conclusion is maintained. We also emphasized the findings of global change factors impact on soil functions are explained by microbial community structure and microbial biomass but not alpha microbial diversity according to your and other reviewers' comments.

3. One major technical problem is that different studies in the dataset used different methods to measure variables related to microbial diversity and functioning. Even the authors used the response ratios (rather than the original values) in this synthesis, this problem could still affect the robustness of the correlations among different studies. For example, the microbial beta diversity is calculated from the community ordination plots by different methods (PCA, RDA, CA, NMDS, PCoA, L313-315), which makes the metric of "beta diversity" hardly comparable among different studies. Also, the variables for the "functioning" were also measured by study-specific method. For example, how the incubations were conducted to measure microbial

respiration and N mineralization, how the assays were conducted to estimate enzyme activities. This method issue should be considered. Overall, I think the core results do not advance the science on microbial diversity and functioning to a significant level, given the limitations listed above.

Response: Thanks. We compared the RRs of microbial beta diversity and community structure to GCFs from different ordination analyses including canonical correspondence analysis (CCA), non-metric multidimensional scaling (NMDS), principal correspondence analysis (PCoA), redundancy analysis (RDA), and principal component analysis (PCA). We found that ordination methods inconsistently influenced the responses of microbial beta diversity and community structure to GCFs and these effects were not significant in most GCFs based on the omnibus test (**Supplementary Fig. 12b, c**). These results suggest that the ordination analyses do not introduce much bias for RRs of microbial beta diversity and community structure in a global analysis. Also please refer to the comments from **Reviewer #2** and our responses to your first comments.

Reviewer #2 (Remarks to the Author):

This manuscript assesses the impact of global changes on soil microbial diversity and functionality. I evaluated this manuscript again. I think the authors did a very thorough job (also including many more studies in their meta-analysis) and answered my queries properly. I think this is an important study. The presented links between microbial biomass and functionality are also appealing. I have mainly minor comments. I must admit that the abstract is not very specific (a bit broad and a bit “vague”) and I feel it is necessary to integrate a few clear conclusions to attract a broader readership and get many more citations of this work (see specific comments for suggestions).

Response: Thank you for your support and invaluable suggestion. We revised the abstract and conclusion to be more specific. Please also refer to the responses below.

Specific comments.

Please make sure all abbreviations are explained in the text (e.g. line 85: “PPT”, “W”, “LUC”, perhaps it is explained or partly explained but I missed it).

Response: Thanks. We confirmed.

Line 105: “Additionally, a significant increase in alpha diversity during the conversions from highly-diverse natural ecosystems to homogeneous agricultural monocultures (Supplementary Fig. 2) implies that changes in microbial alpha diversity are also uncoupled with the shifts in plant alpha diversity.” I think this is a nice observation and this should be mentioned in the abstract.

Response: Thanks. We mentioned it in the summary but not in the abstract due to the limitation of word numbers.

Line 126: “suggesting that rare species are more sensitive to GCs than common species, in agreement study with a previous study (26).” However, that previous study is not a meta-analysis? Right? I thought this is a nice observation and worth to be mentioned in the abstract.

Response: Yes, Reference (26) was a meta-analysis studying on N addition. Here, we found a consistent pattern that rare species are more sensitive to all global change factors than common species, which is mentioned in both abstract and summary as you suggested.

26. Wang, C., Liu, D. & Bai, E. Decreasing soil microbial diversity is associated with decreasing microbial biomass under nitrogen addition. *Soil Biol. Biochem.* 120, 126–133 (2018).

Line 117: “We also found that GCs greatly change microbial community composition regardless of the effects of GCs on microbial diversity mentioned above”. Perhaps also a conclusion that could be suitable for the abstract. This study found that richness is often not strongly responding to GC. However, microbial composition (and the occurrence of rare taxa) changed substantially). I think it is important to mention in the discussion that other microbial community traits such as microbial network structure and characteristics (connectance, keystone taxa, etc.) have also potential (or a better potential) to unravel effects of GC factors on microbial communities.

Response: It is a great idea. Microbial network analysis is a powerful tool to reveal the microbial community structure. However, such analysis needs relatively large sample sizes, and may not be appropriate for our analysis because most of the studies in our datasets had small sample sizes. We added statement associated with community structure in the discussion and conclusion sections.

Figure 5e and 5f: the y-axis for 5e shows “absolute functionality” and the y-axis for 5f shows “functionality”. Please explain the difference.

Response: Thanks. We used the absolute functionality when we explored the relationship between RR of community structure and RR of functionality because greater change in microbial community structure was assumed to result in greater change (increase or decrease) in functionality and the community structure can not be quantified as increasing or decreasing. We clarified it.

Supplement Figure 1: The distribution of the different sampling sites shows that most studies used for this work are from China, the US and some from Europe. From some locations (e.g. Africa, Northern Europe/North Asia, South America, there are hardly (or none) data entries. This reviewer is a European and I am sure that for some of the assessed global change factors, studies have been missed. For instance, I am sure there are more studies that assessed the impact of nitrogen addition or land use change on microbial communities. It is important to write a qualifier about this in the discussion. I think with such a broad study, it is hard to include all studies, so for me the current analysis is absolutely fine. But it is important to mention this because it could affect the conclusion and outcome of the study (if more studies had been included). Also, it is important to state clearly that the data are largely from China, the US and Europe and that there are large gaps on the worldwide map.

Response: We agree with you and may miss some studies. The current dataset, however, was selected and compiled using the literature searching protocol and the criteria described in **Data collection** in **Method** section, and we did find that the studies met our criteria were mainly from China, the US and Europe.

Note from the editor: The report has been modified with additional comments from Reviewer 2 on the new comments from Reviewer 1. The comments from Reviewer 2 are shown with asterisks.

Reviewer 1 comments:

I reviewed the previous version of this manuscript. Overall, I think the massive dataset assembled by the authors contributes to our understanding of the responses of soil microbial communities to global change factors. However, I still have a few serious concerns for the current version.

1. The common way of testing the diversity-functioning relationship is from either observational studies at many sites regionally or globally (but using the same method to quantify both diversity and functions) or experimental studies at single site (but manipulating richness or diversity, even for soil microbes). In both ways, the measurement method should be consistent and comparable among sites or treatments. The rationale in this study is to correlate responses of microbial diversity against responses of soil functions (e.g. C and N mineralization rates, enzyme activities) to global changes. Other factors could confound this correlation, such as changes in plant communities (diversity and productivity) and variations in the methods to quantify both microbial diversity and functioning among various studies. Therefore, I think the correlations reported (e.g. Fig. 5) are very uncertain and not strong enough to support the argument that "... advance the biodiversity-productivity-functionality relationships in microbial ecology..." (L253-254).

REVIEWER 2 RESPONSE: Obviously, it would be better if the same methods had been used. However, the authors compared a control treatment with a GC treatment for each study. Subsequently it is possible to standardize the data and compare many different studies using response ratios. I think this is fine. I think the reviewer is right about the last statement. It is very strong and not really necessary and they could remove it: Something like: "Overall, our findings INDICATE THAT THE RESPONSE OF microbial communities TO GLOBAL CHANGES are fundamentally different from those for macro-communities."

Response: Thanks. Revised.

2. In soil microbial ecology, an increasingly accepted idea is that the physiologically narrow processes (e.g. nitrification) are controlled by microbial diversity, while the physiologically broad processes (e.g. decomposition) may not due to microbial redundancy. Therefore, different functions should be considered against microbial diversity differently. Moreover, microbial biomass can be more important than microbial diversity in controlling function. The relative role of diversity vs. biomass on function should be separated.

REVIEWER 2 RESPONSE: This point could be addressed. I think the reviewer makes a good point here and it is. Probably good to further discuss this issue. I find it very interesting and important (see also my comments below).

Response: Thanks. Revised. Please refer to the response to Reviewer 1.

3. One major technical problem is that different studies in the dataset used different methods to measure variables related to microbial diversity and functioning. Even the authors used the response ratios (rather than the original values) in this synthesis, this problem could still affect the robustness of the correlations among different studies. For example, the microbial beta diversity is calculated from the community ordination plots by different methods (PCA, RDA, CA, NMDS, PCoA, L313-315), which makes the metric of "beta diversity" hardly comparable among different studies. Also, the variables for the "functioning" were also measured by study-specific method. For example, how the incubations were conducted to measure microbial respiration and N mineralization, how the assays were conducted to estimate enzyme activities. This method issue should be considered.

REVIEWER 2 RESPONSE: I think it is correct to use the response ratio (RR) because this makes it possible to compare different studies. Note that the authors used the RR. Various studies (e.g. with biomass data) also use the logRR or lnRR because that can improve the normality of the data (please verify if this is the case for your data-set). Note that some authors used the sampling variances as proposed in Nakagawa et al. to account for sampling uncertainty in each observation (correcting for differences in sample size among studies and the corresponding difference in uncertainty and robustness of the result) – see also Knapp & van der Heijden 2018, Nature Communication, for references.

The RR also enables to automatically standardize across studies that used different methods, so I think the concerns of the reviewer are addressed by the author. I think the supplemental figure 10 shows that only in a few cases diversity measures gave different results in different studies, which clearly indicates that the approach is valid. It probably would be good if the authors clarify in the figure legend of this figure how they then could test whether there was a significant difference between measures for a particular study / observation (e.g. I assume this is because each study had various replications for the control and the GC treatment). I realize now, however, that such a supplemental figure is not shown for the different beta diversity matrixes used (e.g. PCA, RDA, CA, NMDS, PCoA,). This should be included as well and the reviewer correctly points to this (line 382 & 400 actually says it is presented in supplementary figure 10, but only microbial alpha diversity indexes are compared there – so this figure is missing!). Also, the methods used to calculate beta-diversity are more diverse and as such perhaps pooling them all is perhaps more critical compared to the alpha diversity indexes. Therefore, such a figure is necessary and such issues need to be discussed (can also be a discussion for a supplement or in the methods section).

Moreover, figure 5c suggests that alpha diversity is not important for functionality. It is important to clearly state (and discuss) that the change in alpha diversity is negatively correlated with functionality. This does not necessarily say that microbial alpha diversity is not important for functionality because the observations in this meta-analysis are based on changes within experiments, not a global correlation between microbial diversity and functionality (e.g. various

studies show that microbial alpha diversity is positively linked to functionality – Wagg et al. 2014 (PNAS), 2019 (Nature Comm), Jing et al. 2015 (Nature Comm), Delgado-Baquerizo et al. 2020 (NEE)). This issue needs to be carefully discussed. I like the notes about active versus dormant diversity microbes (line 223) and the importance of microbial biomass/microbial composition versus microbial diversity. I think the observations in the figure will lead to discussion and will for sure develop the science in this area. Obviously more work is needed here. As such I appreciate this.

4. Overall, I think the core results do not advance the science on microbial diversity and functioning to a significant level, given the limitations listed above.

REVIEWER 2 RESPONSE: I think this is one of the largest meta-analysis I have seen on this subject and as such I do think it advances the science. The reviewer mainly points to technical issues, which is a matter of debate (see my response above for point 3), the question of novelty/relevance is not criticised. I am not aware of a similar meta-analysis in the literature, but it is always possible I missed something as there is such a huge number of studies coming out.

Response: Thanks.

=====

Reviewer #3 (Remarks to the Author):

Again, I would like to commend the authors for extensive re-analysis and revisions in the manuscript. I think most of technical questions raised by the reviewers have been addressed. Inclusion of functional communities in the manuscript is really appreciated. My main concern regarding framing and presentation of the paper remains.

1. Authors use alpha- diversity liberally to proxy of biodiversity. As authors have mentioned biodiversity includes multiple attributes including, alpha, beta, gamma (etc) diversity. It is important that authors consistently use the term alpha diversity (richness or Shannon diversity) throughout the manuscript to avoid confusion.

Response: Thanks for your insightful comments. Revised.

2. The main finding of this manuscript is that GCs impacts on soil functions are explained by 'microbial composition' (I guess author mean microbial community structure here?) and microbial biomass and not alpha microbial diversity. I believe this will be a good addition in literature if authors can structure manuscript accordingly and not get distracted by Microbial BEF (this manuscript does not address this) and how it is similar or different from plant ecology. Authors have highlighted these findings (please see below comments) in the main text but title, abstract and conclusion do not reflect these key findings.

Response: Thanks. First, we changed the phrase of 'microbial composition' to 'community structure' throughout the manuscript. Second, we emphasized that the findings of global change factors impact on soil functions are explained by microbial community structure and microbial

biomass but not alpha microbial diversity. Third, we consequently avoided making claims about biodiversity-ecosystem function relationships in abstract and conclusion sections.

3. Tittle can be more informative, e.g. " microbial community structure and biomass and not alpha- diversity explain effect of GCs on soil functions.

Response: Thanks. According to editor's suggestion, we revised the title to '*Meta-analysis of the impacts of global change factors on soil microbial diversity and functionality.*'

4. L66- 76. This section is irrelevant- the manuscript is not addressing microbial BEF but impacts of GC- on microbial community attributes (alpha diversity, community structure and biomass) and if these can explain change in soil functions. I therefore suggest to delete this section.

Response: Thanks for your suggestion. Indeed, we did not find that global change factors induced loss of microbial alpha diversity would negatively influence microbial functionality. But we think the decoupled microbial alpha diversity and functionality under several global change factors are also an important finding. According to your suggestions, we avoided making claims about biodiversity-ecosystem function relationships in abstract and conclusion, but we kept this research question.

5. In my opinion, following are key findings are (1) GCs greatly change microbial community (L116-122) and it explains significant ($R^2 = 0.58$) variation in soil functions; (2) microbial biomass positively correlated with soil functions ($R^2 = 0.51$) (L227-229) and biomass of specialised community are strongly linked to rate of specialised functions (L240-243). Currently these important findings are hidden in the main text and in my opinion, these should be reflected strongly in title, abstract and conclusion sections.

Response: Thanks. We emphasized these findings in abstract and conclusion sections.

We hope that you find our revision satisfactory. Thank you very much!